# Pharmacological blockade of TEAD–YAP reveals its therapeutic limitation in cancer cells

Yang Sun[1,6,7] ✉, Lu Hu [1,7], Zhipeng Tao [1,7], Gopala K. Jarugumilli[1], Hannah Erb[1], Alka Singh[2], Qi Li[2], Jennifer L. Cotton [2], Patricia Greninger [3], Regina K. Egan[3], Y. Tony Ip [4], Cyril H. Benes[3], Jianwei Che[5], Junhao Mao[2] ✉ & Xu Wu [1] ✉

Targeting TEAD autopalmitoylation has been proposed as a therapeutic approach for YAP-dependent cancers. Here we show that TEAD palmitoylation inhibitor MGH-CP1 and analogues block cancer cell "stemness", organ overgrowth and tumor initiation in vitro and in vivo. MGH-CP1 sensitivity correlates significantly with YAP-dependency in a large panel of cancer cell lines. However, TEAD inhibition or YAP/TAZ knockdown leads to transient inhibition of cell cycle progression without inducing cell death, undermining their potential therapeutic utilities. We further reveal that TEAD inhibition or YAP/TAZ silencing leads to VGLL3-mediated transcriptional activation of SOX4/PI3K/AKT signaling axis, which contributes to cancer cell survival and confers therapeutic resistance to TEAD inhibitors. Consistently, combination of TEAD and AKT inhibitors exhibits strong synergy in inducing cancer cell death. Our work characterizes the therapeutic opportunities and limitations of TEAD palmitoylation inhibitors in cancers, and uncovers an intrinsic molecular mechanism, which confers potential therapeutic resistance.

Hippo signaling plays a critical role in development, regeneration and tumor suppression[1,2] by inhibiting the transcriptional co-activators YAP and TAZ through a kinase-mediated phosphorylation cascade[3]. Inactivation of upstream Hippo pathway components, including SAV1, NF2/merlin, Mst1/2 and Lats1/2, results in YAP and TAZ nuclear localization and binding to TEA/TEF-domain transcription factors (TEAD1–4) to mediate target gene expression[4,5]. TEAD–YAP complex regulates cell proliferation, survival, epithelial-to-mesenchymal transition (EMT)[6–8], and immune evasion of cancers[9,10], playing a critical role in tumorigenesis. Consistently, YAP and TAZ are frequently amplified or hyper-activated in a wide range of human cancers, including breast, ovarian, lung, head and neck, liver and colon cancers[7]. In addition, YAP activation promotes the development of "persister cells" and confers resistance to targeted therapies[11,12]. Therefore, inhibition of TEAD–YAP complex has been proposed as a promising therapeutic approach for a broad range of cancers[13].

Post-translational protein S-palmitoylation attaches 16-carbon palmitoyl group to the cysteine (Cys) residue with a thioester linkage[14]. Dynamic palmitoylation regulates membrane localization, trafficking, and co-factor binding of many proteins[15]. We and others have previously reported that TEADs undergo enzyme-independent autopalmitoylation under physiological conditions at conserved Cys residue[16,17]. Crystal structures reveal a deep, hydrophobic pocket in

[1]Cutaneous Biology Research Center, Massachusetts General Hospital, Harvard Medical School, Charlestown, Massachusetts, MA, USA. [2]Department of Molecular, Cell and Cancer Biology, University of Massachusetts Chan Medical School, Worcester, Massachusetts, MA, USA. [3]Massachusetts General Hospital Cancer Center, and Department of Medicine, Harvard Medical School, Charlestown, Massachusetts, MA, USA. [4]Program in Molecular Medicine, University of Massachusetts Chan Medical School, Worcester, Massachusetts, MA, USA. [5]Department of Cancer Biology, Dana Farber Cancer Institute, and Harvard Medical School, Boston, Massachusetts, MA, USA. [6]Present address: Cancer Institute, Xuzhou Medical University, Xuzhou, Jiangsu, China. [7]These authors contributed equally: Yang Sun, Lu Hu, Zhipeng Tao. ✉e-mail: yangsun@xzhmu.edu.cn; Junhao.Mao@umassmed.edu; xwu@cbrc2.mgh.harvard.edu

TEADs to accommodate palmitate binding, which allosterically regulates TEAD–YAP interaction and TEAD transcriptional activities[17]. Loss of TEAD palmitoylation inhibits TEAD–YAP activities in vitro and in vivo[16], and the lipid-binding pocket could also accommodate small molecule binding[18], suggesting that TEADs are potentially druggable through inhibiting their "autopalmitoylation" activities. Such discovery led to quick identifications of several reversible and irreversible TEAD palmitoylation modulators that bind to the lipid-binding pocket. Several covalent inhibitors were reported to competitively alkylate the Cys residue located at the opening of the pocket, resulting in TEAD–YAP dissociation[19–22]. Interestingly, some small molecules bound to the same site could activate TEAD transcriptional activities[23], suggesting that the lipid-binding pocket is an allosteric site modulating TEADs' functions. We recently reported MGH-CP1 as a reversible small molecule inhibitor of TEAD autopalmitoylation, which inhibits TEAD functions in intestinal stem cells in vitro and in vivo[24]. However, the majority of known TEAD inhibitors only show antiproliferative activities in a few cancer cell lines, such as NF2-mutant mesothelioma, and detailed analysis of TEAD palmitoylation inhibitors in a broad range of YAP-dependent cancer models is still lacking.

In this work, we carry out medicinal chemistry modifications of MGH-CP1 and synthesize a series of its analogues to understand the structure-activity relationship (SAR). We confirm that MGH-CP1 and its analogues are specific TEAD inhibitors in cancer cells, which show a significant overlap with YAP/TAZ knockdown in transcriptomic regulation. TEAD inhibition markedly suppresses cancer cell "stemness", migration, YAP-dependent organ overgrowth and tumor initiation in vitro and in vivo. We find that the sensitivity to MGH-CP1 significantly correlates with YAP-dependent cell growth. However, TEAD–YAP blockade by MGH-CP1 or YAP/TAZ knockdown only leads to transient cell proliferation stasis in YAP-dependent cancer cell lines, without cell death induction. To understand the mechanisms, we compare transcriptomic output of MGH-CP1 treatment and YAP/TAZ knockdown, and find that a subset of genes, including *PIK3C2B* and *SOX4*, are commonly and strongly induced by TEAD–YAP blockade, which might constitute a feedback regulation to limit the efficacy. Interestingly, we find that VGLL3 could regulate transcriptional activation of these target genes upon TEAD–YAP blockade, suggesting that alternative activation of TEAD–VGLL complex might compromise the effects of TEAD–YAP inhibition. Consistently, MGH-CP1 and several other TEAD inhibitors strongly induce AKT activation. Therefore, VGLL3-mediated activation of SOX4/PIK3C2B/AKT axis might be part of the mechanisms that limit TEAD inhibitors' anti-tumor activities. Indeed, combination of MGH-CP1 with an AKT inhibitor shows significant synergy and promotes cancer cell death. Taken together, our studies reveal the therapeutic opportunities and limitations of pharmacological blockade of TEAD palmitoylation and uncover the functions of TEAD–VGLL3 complex in cancers. Such work also suggests that rationalized combination of TEAD inhibitors with AKT inhibitors might provide more effective anti-cancer therapies.

## Results

### TEAD palmitoylation inhibitors could have different functional output in blocking TEAD-YAP association and transcriptional activities

Previously, we and others reported that TEADs undergo autopalmitoylation, playing a critical role in regulating their transcriptional activities[16,17]. Subsequently, we have identified a small molecule compound, MGH-CP1, as a pan-inhibitor of TEADs by binding to the lipid-binding pocket and inhibiting TEAD–YAP mediated stem cell functions in vitro and in vivo[24].

To gain insights into the structure-activity relationship (SAR) of MGH-CP1 series of compounds, we rationally designed and synthesized analogues of MGH-CP1 and tested these compounds in TEAD autopalmitoylation assay in vitro (Fig. 1a, b, Supplementary Fig. 1a).

Among the analogues, we found that replacing the adamantyl substituent with less hydrophobic moieties substantially decreases the potency of the inhibitors (MGH-CP2 and MGH-CP9), consistent with the extraordinary hydrophobicity inside the lipid-binding pocket. Installation of an *ortho*-bromo group at the phenyl ring leads to >10-fold improvement of potency in TEAD2 biochemical palmitoylation assay (MGH-CP12 with an IC$_{50}$ of 0.106 μM). Oxidation of thioether to sulfoxide in MGH-CP1 abolishes its potency, while oxidation to sulfone is tolerated as demonstrated by MGH-CP25-1 and MGH-CP25. Interestingly, increasing the length of the linker between triazole and aniline by one methylene slightly improves the activities, and extending to two methylene groups abolishes the activities (MGH-CP27 and MGH-CP34). In addition, methylation of triazole N-H decreases the activities by 2-fold (MGH-CP8). Replacing the triazole with thiazole ring largely maintains the potency (MGH-CP28 with an IC$_{50}$ of 0.617 μM). To gain the insights of SAR, co-crystal structure of TEAD2 protein bound with MGH-CP1 was used to generate docking grid for MGH-CP12 and MGH-CP34 (Fig. 1c). Interestingly, the *ortho*-bromo group at the phenyl ring stabilizes the hydrophobic interactions between MGH-CP12 and residues Phe 233, 302 and 428 of TEAD2 protein, enhancing its binding affinity. In MGH-CP34 structure, the two additional methylene groups are not beneficial to maintaining the compact structure of the inhibitor when it binds to TEAD2, resulting in loss of potency (Fig. 1c). As MGH-CP12 has shown improved in vitro TEAD2 inhibitory activities, we tested MGH-CP1 and MGH-CP12 in additional assays. Although MGH-CP12 inhibits TEAD1 palmitoylation in cells more potently than MGH-CP1 (Fig. 1d), MGH-CP12 behaves similarly to MGH-CP1 in the TEAD4 biochemical palmitoylation assay (IC$_{50}$ of 0.852 μM for MGH-CP12 and IC$_{50}$ of 0.672 μM for MGH-CP1) (Supplementary Fig. 1b). Consistently, both MGH-CP1 and MGH-CP12 have similar activities in endogenous pan-TEAD palmitoylation in HEK293A cells (Supplementary Fig. 1c), suggesting that MGH-CP12 is a more potent TEAD1/2 inhibitor than MGH-CP1, but has similar activities to MGH-CP1 as a pan-TEAD inhibitor.

We further tested MGH-CP1, CP12, CP27 and CP28 in blocking TEAD–YAP interaction using a Gal4-TEAD reporter assay[4]. We found that MGH-CP12 is more potent than MGH-CP1 in blocking the interaction between Gal4-TEAD1 or Gal4-TEAD2 and YAP (Fig. 1e), with an IC$_{50}$ of 0.302 μM in Gal4-TEAD2-YAP binding assay (Supplementary Fig. 1d). Interestingly, MGH-CP27 and CP28 only showed weak activity in this assay, although both compounds block TEAD2 palmitoylation potently in vitro (Fig. 1e). As several compounds binding to this site might only block TEAD palmitoylation, but do not inhibit TEAD–YAP interaction strongly or even activate TEAD–YAP activity in some cases, the lipid-binding pocket functions as an allosteric site to regulate TEAD–YAP association[3,23]. The detailed mechanisms of how palmitate or inhibitor binding allosterically regulates TEAD functions are still elusive. Nevertheless, functional assays would be needed to distinguish TEAD binders, activators, and inhibitors.

In addition, we found that MGH-CP1 inhibits TEAD–TAZ interaction in a co-immunoprecipitation experiment (Supplementary Fig. 1e) but does not inhibit TEAD-DNA binding in a DNA pull-down assay in cells[25] (Supplementary Fig. 1f). We did not observe significant TEAD protein level decrease upon inhibitor treatment, suggesting that palmitoylation inhibition might not have strong effects in modulating TEAD stability. These results are also consistent with other reported TEAD palmitoylation inhibitors, such as VT103, which do not decrease TEAD protein levels dramatically[26].

Consistently, we found that MGH-CP1 and MGH-CP12 could potently inhibit TEAD–YAP transcriptional activities in TEAD-binding element driven luciferase reporter assays with IC$_{50}$s of 1.68 μM and 0.91 μM, respectively (Fig. 1f and Supplementary Fig. 1g), and suppress the expression of the TEAD–YAP target genes similarly (*CTGF*, *Cyr61* and *ANKRD1*) in cancer cell lines in qPCR assays (Fig. 1g). In addition, MGH-CP1 and MGH-CP12 significantly inhibit tumor sphere

formation in YAP-dependent liver cancer cell line (Huh7 cells) in a dose-dependent manner with $IC_{50}$ values of 0.72 μM and 0.26 μM, respectively (Fig. 1h and Supplementary Fig. 1f), suggesting that the tumor sphere inhibition is correlated with its TEAD1 or TEAD2 inhibitory activities. Both compounds also potently inhibit the secondary tumor sphere formation, suggesting that TEAD inhibition indeed could block "stemness" features of cancer cells (Fig. 1h and Supplementary Fig. 1f). Furthermore, knockdown of YAP/TAZ by siRNAs confirmed that HUTU80 cells are more dependent on YAP/TAZ than HCT116 cells[27], and their sensitivities to MGH-CP1 and MGH-CP12 inhibition in 2D and 3D culture conditions correlates with the dependency (Fig. 1i).

## MGH-CP1 and YAP/TAZ knockdown show significant overlap in modulating transcriptomic output in cancer cell lines in RNA-seq analysis

To analyze the transcriptomic output of MGH-CP1 treatment and YAP/TAZ knockdown, we performed Fisher's Exact Test using a newly generated and improved RNA-seq dataset of MGH-CP1 (GSE177052) versus YAP/TAZ knockdown (GSE102407) in MDA-MB-231 cells (https://www.ncbi.nlm.nih.gov/geo/query/acc.cgi?acc=GSE177052 and https://www.ncbi.nlm.nih.gov/geo/query/acc.cgi?acc=GSE102407)[28]. The results show significant overlap of MGH-CP1 and YAP/TAZ knockdown in the cancer cell line with transcriptome-wide analysis (Fig. 2a). A principal component analysis (PCA) was performed to provide a quick and global comparison of relative similarity of transcriptomic output as well (Supplementary Fig. 2a). Our analysis shows that a core expression signature from PC1 (First principal component) could clearly segregate control groups (DMSO and siControl) and treatment groups (MGH-CP1 and siYAP/TAZ) (Supplementary Fig. 2a). In addition, we observe a strong correlation of gene fold changes between MGH-CP1 treatment and YAP/TAZ knockdown (Pearson r = 0.6271, P < 0.0001) (Supplementary Fig. 2b). These analyses strongly support that MGH-CP1 shares significant overlap with YAP/TAZ siRNA in modulating gene expression and suggest that MGH-CP1 is a specific TEAD−YAP/TAZ inhibitor. We further performed The Gene Set Enrichment Analysis (GSEA) comparing MGH-CP1 treatment with a defined TEAD−YAP/TAZ target gene signature[29]. MGH-CP1 strongly inhibits TEAD− YAP/TAZ target genes with the normalized enrichment score at −3.4 (Fig. 2b). Taken together, these data demonstrate that MGH-CP1 specifically and effectively blocks TEAD−YAP/TAZ mediated transcriptional outcome in cancer cells.

## TEAD inhibitors suppress YAP-dependent transformation, organ overgrowth and tumorigenesis in vitro and in vivo

To evaluate the effects of TEAD inhibitors on tumor cell self-renewal, we tested tumor sphere formation of MCF10A cells transformed with YAP wild type or YAP S127A in 3-dimensional culture. YAP expression, especially YAP S127A mutant, could substantially promote formation of tumor spheres, supporting the notion that YAP plays a critical role in transformation (Fig. 2c). We observed that MGH-CP1 reduces the tumor sphere number in a dose-dependent manner (Fig. 2c, d). Similarly, MGH-CP1 significantly inhibits anchorage-dependent colony formation in Huh7, MDA-MB-231 and HCT116 cells (Supplementary Fig. 3a), confirming that TEAD inhibition could block YAP-dependent transformation and colony formation in multiple cell lines.

YAP/TAZ has been reported as an essential factor for persistent migration[30]. To evaluate whether TEAD inhibitor could be effective in blocking tumor cell migration, we performed scratch assay and transwell assays in MDA-MB-231 and Huh7 cells. We found that MGH-CP1 significantly suppresses tumor cell migration (Supplementary Fig. 3b, c, e). In addition, MGH-CP1 treatment significantly decreases the expression of cell migration related gene expression (MACF1, PARD3, PHLDB2, ABL2, CNN3 and DOCK5), examined by qPCR[31] (Supplementary Fig. 3d).

To further examine the effect of MGH-CP1 in vivo in genetically engineered YAP-dependent models, we carried out Lats1/2 double-knockout in adult mouse liver through AAV-Cre recombinase delivery. As expected, Lats1/2 deletion results in rapid and severe hepatomegaly within two weeks. MGH-CP1 treatment (75 mg/kg, once daily) via intraperitoneal injection is well-tolerated and does not affect liver weight in control animals, while significantly inhibiting the overgrowth phenotype in the liver of Lats1/2 double-knockout mice. MGH-CP1 treatment almost completely inhibited liver overgrowth induced by Lats1/2 deletion, and the liver sizes of treated animals are not statistically different from the control normal wild type mice (Fig. 2e, f). Histology and immunohistochemistry analyses reveal that, in the Lats1/2 knockout animals, MGH-CP1 treatment markedly decreases the size of the proliferative clusters and the number of the mitotic cells in mutant animals, measured by phosphorylated-H3 staining (Fig. 2g, h). Interestingly, we noticed that MGH-CP1 treatment does not abolish, but appears to reduce, YAP/TAZ nuclear localization within individual proliferative clusters, probably due to MGH-CP1 inhibition of TEAD−YAP/TAZ interaction, which might lead to weakened nuclear retention of YAP and TAZ proteins.

We further tested MGH-CP1 in subcutaneous xenograft model of YAP-dependent Huh7 cell line. Consistently, MGH-CP1 at 50 mg/kg once daily dose via intraperitoneal injection is well-tolerated in animals, and significantly inhibits the growth of Huh7 xenograft tumors by 43% (Fig. 2i and Supplementary Fig. 4a, b). Interestingly, we observed significant inhibition of TEAD−YAP target genes, including Cyr61, CTGF and ANKRD1, confirming that TEAD−YAP complex is effectively inhibited (Fig. 2j). Uveal melanoma (UM) with GNAQ/GNA11 mutations has been shown as YAP-dependent[32–34]. We tested MGH-CP1 in UM 92.1 cell line tumor growth in vivo and observed that MGH-CP1 significantly blocks UM tumor growth (Supplementary Fig. 4c, d). Importantly, TEAD−YAP target genes Cyr61 and CTGF are significantly inhibited in treated tumors (Supplementary Fig. 4f). In addition, MGH-CP1 (25 mg/kg, 50 mg/kg and 75 mg/kg intraperitoneal injection) treatment significantly inhibits tumor initiation in vivo in Huh7 and MDA-MB-231 xenograft models (Fig. 2k and Supplementary Fig 4g–i). Treatment of cells with MGH-CP1 or MGH-CP12 in vitro for 24 or 48 h before implantation, also dramatically inhibits Huh7 and MDA-MB-231 tumor initiation (Fig. 2l and Supplementary Fig. 4j–l). Taken together, inhibition of TEAD could inhibit YAP-dependent tumor initiation and growth in vivo.

## MGH-CP1 sensitivity correlates with YAP-dependency in a large-scale cancer cell lines profiling

YAP activation has been implicated as a critical oncogenic event in a broad range of cancers[35]. To evaluate whether TEAD inhibition could be a potential cancer therapy strategy, we evaluated the growth inhibition sensitivity of a large panel of human cancer cell lines (360 cell lines) upon MGH-CP1 treatment (Fig. 3a−c)[36]. This dataset was then intersected with the Cancer Dependence Map, in which genome-wide loss-of-function screens through shRNA knockdown or CRISPR-Cas9 knockout were performed to determine the dependence of individual genes in a large collection of human cancer cell lines (https://depmap.org/portal/)[37,38]. We performed the receiver operating characteristic (ROC) curve analysis to examine whether the cells with YAP-dependency would be sensitive to MGH-CP1 treatment (defined as $IC_{50}$ less than 10 μM, while MGH-CP1 insensitive cell lines have $IC_{50}$ values higher than 20 μM). The results showed that YAP-dependency could predict MGH-CP1 sensitivity (Fig. 3d). Pearson correlation analysis between YAP-dependency score from the Dep-Map dataset and $IC_{50}$ values of MGH-CP1 were then carried out. In colon, non-small cell lung, and ovarian cancer cell lines, we observed significant correlations between MGH-CP1 sensitivity with YAP-dependency (Fig. 3e–g). Uveal melanoma cells harboring GNAQ/GNA11 mutations have been shown as YAP-dependent cancer cell

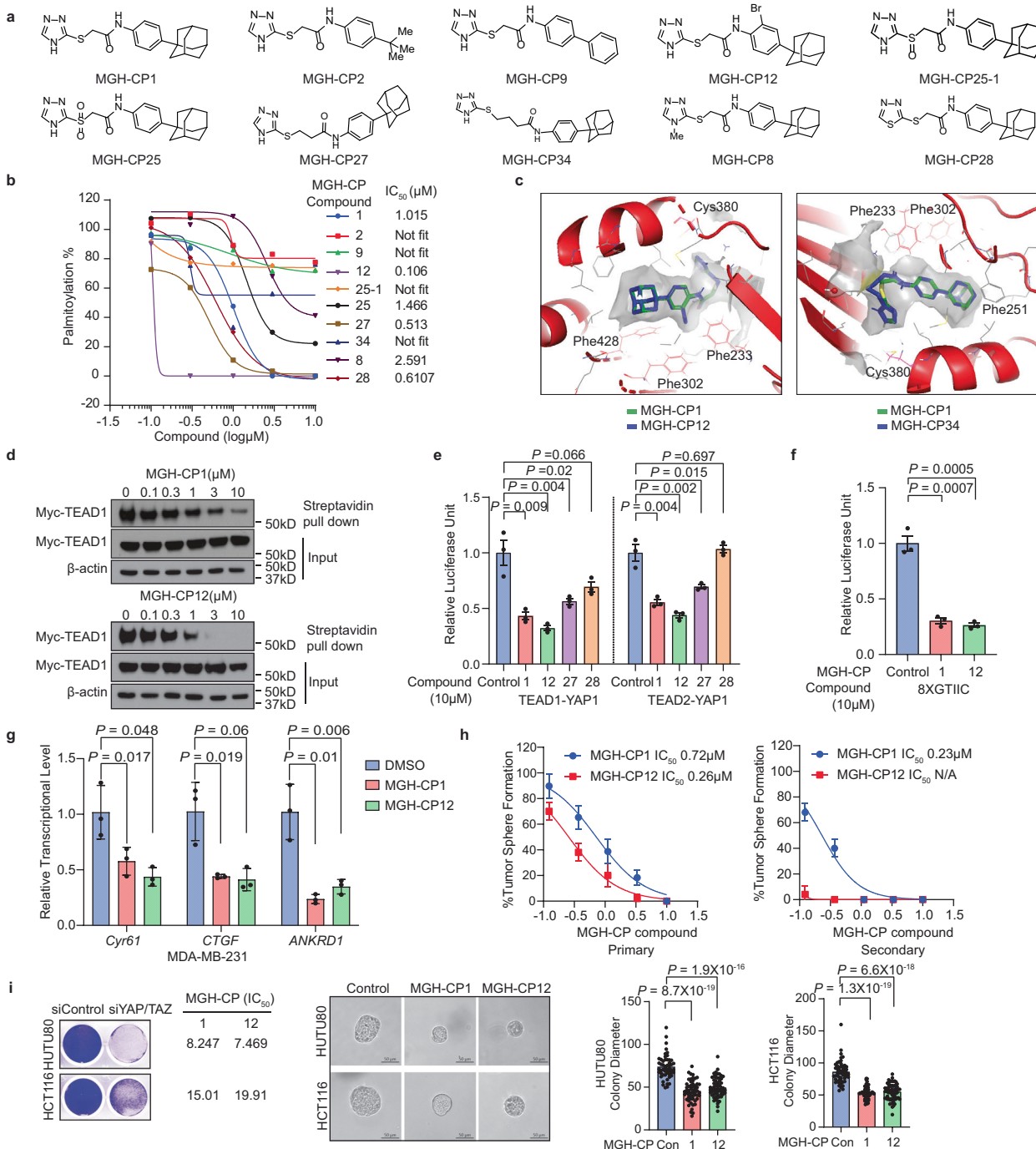

**Fig. 1 | MGH-CP1 and analogues are palmitoylation inhibitors with different outcomes in blocking TEAD–YAP activities. a** Structures of MGH-CP1 and its analogues. **b** Inhibition of TEAD2 YAP-binding domain (YBD) in vitro autopalmitoylation by MGH-CP compounds. **c** The MGH-CP12 and MGH-CP34 were docked into MGH-CP1 binding pocket of TEAD2 YAP binding domain. **d** Cell-based palmitoylation assay using metabolic probe alkyne palmitic acid labeled protein. HEK293A cells were treated with MGH-CP1 and MGH-CP12 separately. **e** Inhibition of TEAD1/2 and YAP binding in Gal4-TEAD1/2-YAP binding reporter assay by MGH-CP1, 12, 27 and 28 (*n* = 3 biological repeats). **f** MGH-CP1 and 12 inhibit TEAD-binding element–driven luciferase reporter (8xGTIIC-luciferase) (*n* = 3 biological repeats). **g** MDA-MB-231 cells were treated with MGH-CP1 and MGH-CP12. Transcriptional levels of *Cyr61, CTGF* and *ANKRD1* were determined using qPCR (n = 3 biological repeats). **h** Primary and secondary Huh7 tumor spheres treated with MGH-CP1 and MGH-CP12 with indicated concentrations. Dose response curves show MGH-CP1 and MGH-CP12 in Huh7 tumor sphere formation assay (*n* = 3 biological repeats). **i** Representative images of HCT116 and HUTU80 cells with knockdown of YAP/TAZ, compared to MGH-CP1 and MGH-CP12 IC$_{50}$ values, and representative images in 3D tumor cultures with MGH-CP1 and MGH-CP12 treatment in HUTU80 and HCT116 cells. The 3D colonies were measured by diameter at day 4 with compound treatment (*n* = 51, 53, 59 colonies for control, CP1 and CP12, respectively in HUTU80 cells, *n* = 52, 58, 66 colonies for control, CP1 and CP12, respectively in HCT116 cells. Scale bar, 50 μm. Data are represented as mean ± S.E.M. *P* values were determined using two-tailed *t*-tests. Source data are provided as a Source Data file.

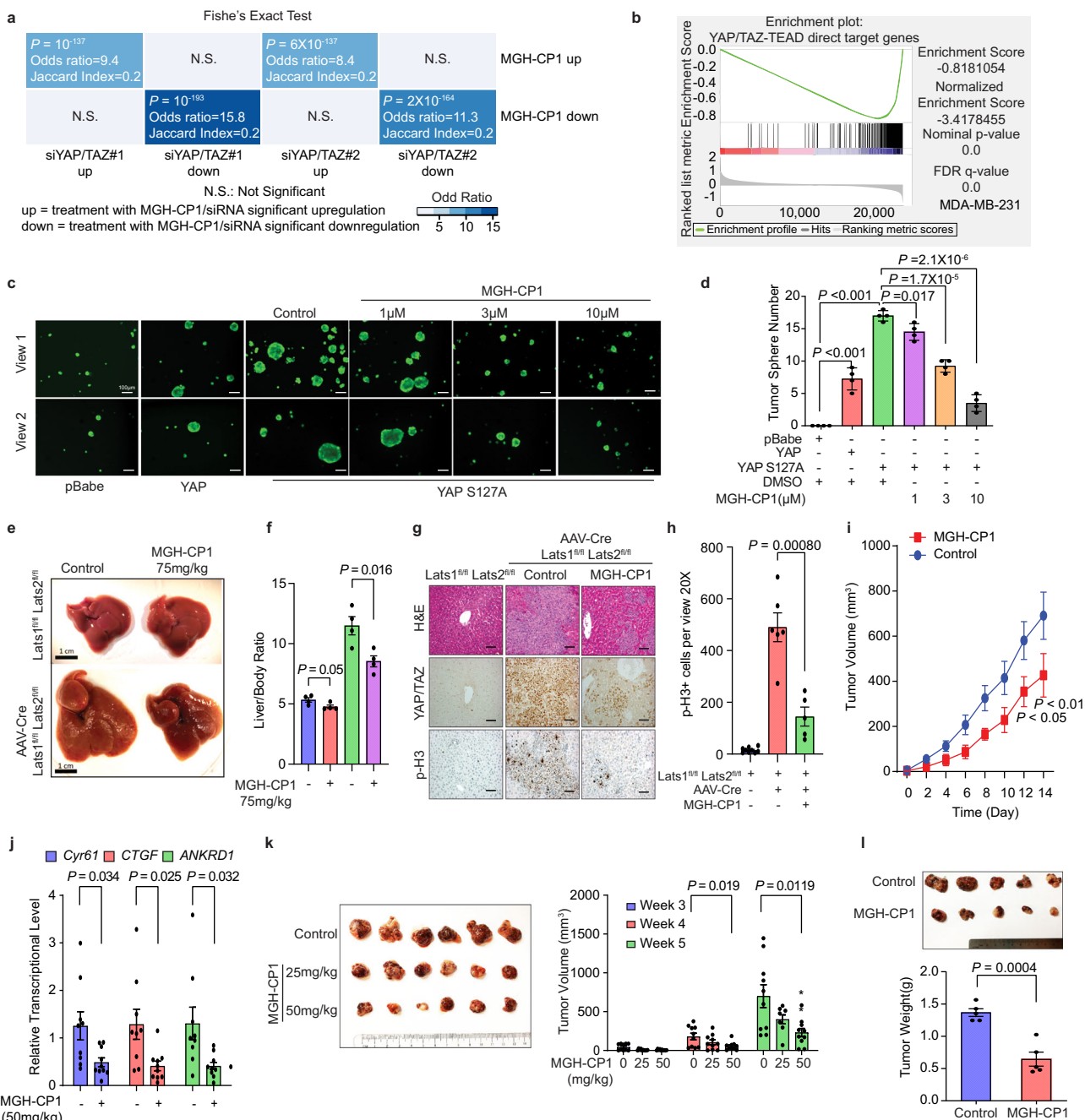

**Fig. 2 | MGH-CP1 treatment overlaps with YAP/TAZ suppression in transcriptional output and inhibits YAP-dependent transformation, organ enlargement, and tumor initiation. a** Fisher's Exact Test using RNA-seq datasets of MGH-CP1 (GSE177052) versus YAP/TAZ knockdown (GSE102407) in MDA-MB-231 cells. Two tailed *P* values are shown. **b** Gene set enrichment analysis of YAP/TAZ-TEAD target gene signature upon MGH-CP1 treatment. **c** Tumor sphere formation assay of MCF10A cells transduced with YAP WT or S127A mutation. Indicated doses of MGH-CP1 were applied. Tumor sphere numbers were counted in **d** (*n* = 4 biological repeats). Scale bar, 100 μm. **e** MGH-CP1 inhibits liver enlargement induced by liver specific *Lats1/Lats2* deletion in vivo. Scale bar, 1 cm. **f** Quantification of relative liver weight of control or MGH-CP1-treated *Lats1$^{fl/fl}$;Last2$^{fl/fl}$* mice with control or Lentiviral-Cre delivery (*n* = 4 mice). **g** Representative images of histology and IHC of YAP/TAZ and phospho-histone H3 (p-H3) in the liver. Scale bar, 50 μm. (Images were chosen from *n* = 9, 6, 5 histology fields for Control, *Lats1/Lats2* double knockout and *Lats1/Lats2* double knockout with MGH-CP1, respectively)

**h** Percentage of phospho-histone H3 positive cells in the liver of control and *Lats1/Lats2* double knockout mice treated with MGH-CP1 (*n* = 9, 6, 5 histology fields for Control, *Lats1/Lats2* double knockout and *Lats1/Lats2* double knockout with MGH-CP1, respectively). **i** Xenograft tumor volumes of Huh7 cells inoculated in SCID mice, and then treated with vehicle control or MGH-CP1 (*n* = 9, 10 tumors for control and MGH-CP1, respectively). **j** Relative mRNA levels of *CTGF* and *Cyr61* were determined using qPCR in xenograft tumors. All the mRNA levels were normalized to *18S rRNA* (*n* = 9, 10 tumors for control and MGH-CP1, respectively). **k** The image of xenograft tumors are shown for tumor initiation. Tumor volumes were determined (*n* = 10 tumors). **l** The images of xenograft tumors pre-treated with Control or MGH-CP1 (10 μM for 24 h) ex vivo, before inoculation into animals. Tumor weights were measured (*n* = 5 tumors). Data are represented as mean ± S.E.M. *P* values were determined using two-tailed *t*-tests. Source data are provided as a Source Data file.

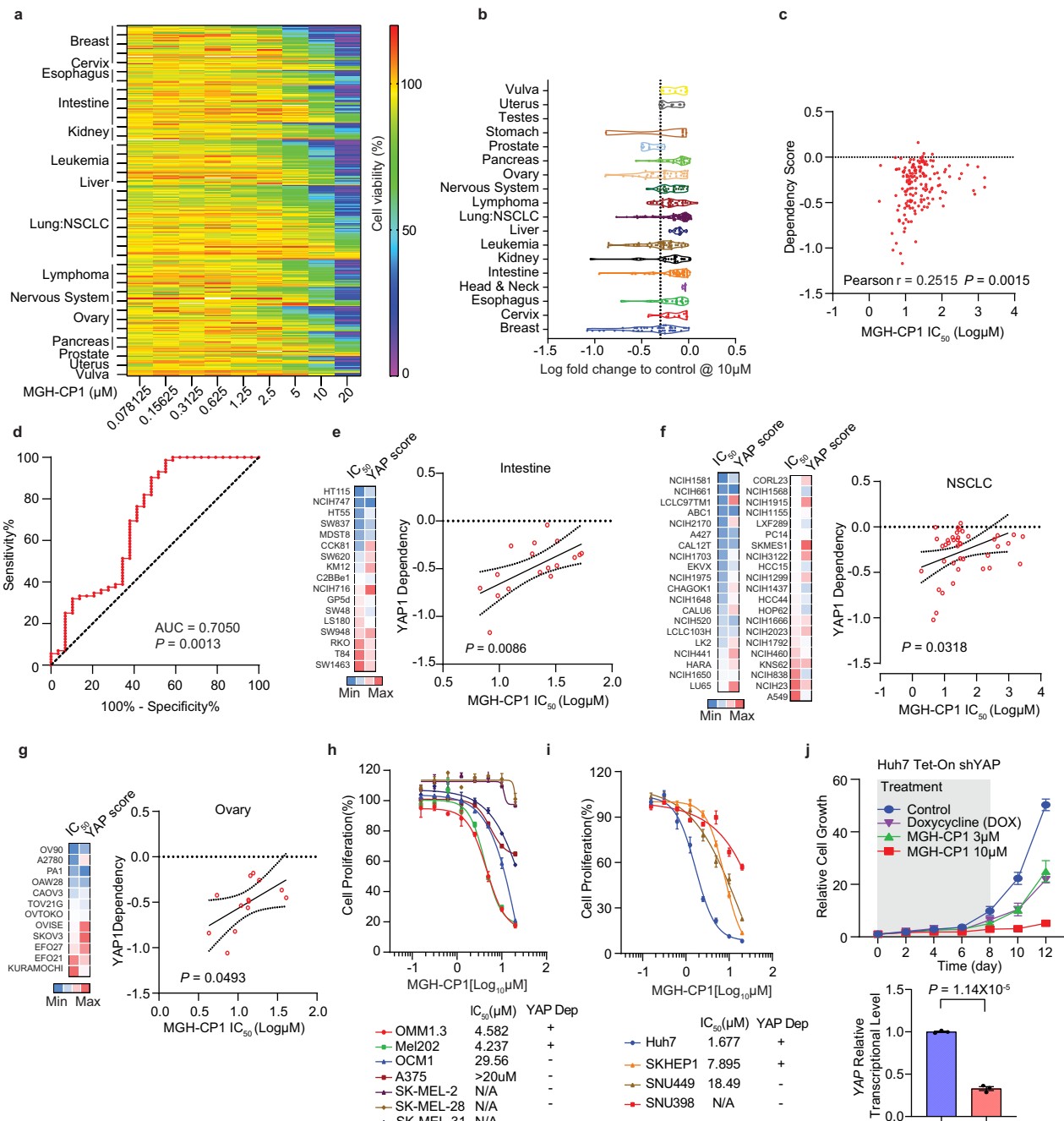

**Fig. 3 | Cancer cell growth sensitivity to MGH-CP1 correlates with YAP-dependency in a large-scale cancer cell line profiling. a** Heatmap shows viability through color-coding as percentage of cell viability normalized to vehicle control in tumor cells treated with serial concentrations of MGH-CP1. **b** Violin plot shows viability fold change by MGH-CP1 treatment relative to vehicle control in different type of tumor cells. **c** A receiver operating characteristic curve shows YAP-dependency of MGH-CP1 sensitive tumor cells (MGH-CP1 $IC_{50}$ < 10 μM) relative to insensitive tumor cells (MGH-CP1 $IC_{50}$ > 20 μM). **d** Pearson correlation of MGH-CP1 $IC_{50}$ against YAP dependency score in a large panel of cancer cell lines. The cell line proliferation profiling in the presence of MGH-CP1 and intersecting with cancer cell dependency map showed significant correlation of YAP-dependency score with MGH-CP1 sensitivity ($IC_{50}$ values) in colon cancer cell lines (**e**), non-small-cell lung

carcinoma (NSCLC) cancer cell lines (**f**) and ovarian cancer cell lines (**g**). **h** MGH-CP1 inhibits the proliferation of YAP-dependent, but not of YAP-independent liver cancer cell lines ($n$ = 3 biological repeats). **i** MGH-CP1 inhibits the proliferation of YAP-dependent GNAQ mutant uveal melanoma cells compared to BRAF or NRAS mutant uveal melanoma and melanoma cell lines ($n$ = 6 biological repeats for Huh7, SK-HEP-1 and SNU449, $n$ = 3 biological repeats for SNU398). **j** Tet-on inducible shRNA knockdown of YAP or MGH-CP1 treatment in Huh7 cells was administrated for 8 days. Doxycycline or MGH-CP1 was withdrawn after 8 days, and cell proliferation is monitored for additional 4 days. YAP knockdown efficiency after doxycycline treatment was evaluated by qPCR after 48 h ($n$ = 3 biological repeats). DOX, Doxycycline. Data are represented as mean ± S.E.M. $P$ values were determined using two-tailed $t$-tests. Source data are provided as a Source Data file.

lines[33,34]. Indeed, we found that MGH-CP1 markedly inhibits the growth of OMM1.3 (GNAQ $^{mt}$) and Mel202 (GNAQ$^{mt}$) with $IC_{50}$ around 4 μM but has little anti-proliferative effects on YAP-independent OCM1 (BRAF$^{mt}$), and cutaneous melanoma cell lines A375 (BRAF$^{mt}$),

SKMEL2 (NRASmt), SK-MEL28 (BRAF$^{mt}$) and SK-MEL31(RAS/BRAF$^{wt}$) cells with $IC_{50}$ > 30 μM (Fig. 3h). In reconfirmation experiments, we found that the previously reported YAP-dependent liver cancer cell lines, Huh7 and SKHEP1, are sensitive responders to MGH-CP1, while

YAP-independent SNU448 and SNU398 liver cancer cells are not sensitive to MGH-CP1[39] (Fig. 3i).

Interestingly, our results showed that TEAD−YAP blockade only inhibits cancer cell proliferation, without significant induction of cell death. To further evaluate these effects, we generated doxycycline-induced stable knockdown of YAP in Huh7 cells, which leads to inhibition of cell proliferation. However, cell proliferation rebound after withdrawal of doxycycline, and similar transient inhibition of cell growth is also observed by treatment and withdrawal of MGH-CP1 (Fig. 3j). Therefore, TEAD−YAP blockade might only delay cell cycle progression, consistent with the report that TEAD−YAP upregulates transcription of cell cycle promoting genes[40]. We further carried out cell cycle analysis, and found that cells were transiently stranded at G1 and G2/M stage in the presence of MGH-CP1, compared to DMSO control in Huh7 cells (Supplementary Fig. 5a, b).

## TEAD−YAP/TAZ blockade promotes VGLL3-mediated transcriptional activation of *PIK3C2B* and *SOX4*

Although TEAD−YAP/TAZ complex has been implicated as a driver and addicted oncogenic factor in many cancers, our results suggest that TEAD−YAP blockade might only lead to transient and moderate anti-proliferation effects, undermining the effectiveness of targeting TEAD−YAP in cancers. We hypothesize that certain feedback regulations upon TEAD−YAP blockade might promote cell survival, resulting in the lack of cell death induction. TEAD−YAP/TAZ transcriptional complex was known to play critical roles in inducing the expressions of a myriad of target genes. However, the transcriptional repressor target genes by TEAD−YAP/TAZ have not been thoroughly studied. In RNA-seq analysis of MGH-CP1 treated cells, we found that 2326 genes are up-regulated, comparable to the number of genes suppressed by YAP/TAZ (Fig. 4a). We compared this dataset with publicly available RNA-seq datasets with YAP/TAZ siRNA knockdown in 5 different cell lines and defined a subset of 51 genes as commonly and significantly upregulated in all these cells upon TEAD−YAP/TAZ blockade (Fig. 4b)[28,41–44]. The heatmap of these 51 genes upon MGH-CP1 treatment is shown in Fig. 4c and listed in Supplementary Table 1. Interestingly, *PIK3C2B* and *SOX4* are among the top genes suppressed by YAP/TAZ activation and induced by TEAD−YAP blockade (MGH-CP1 and YAP/TAZ knockdown). Consistently, the expression levels of *PIK3C2B* negatively and significantly correlate with YAP-induced target gene signature in Pearson correlation analysis (Fig. 4d). These results suggest that *PIK3C2B* and *SOX4* are induced upon TEAD−YAP blockade. It has been reported previously that TEAD−YAP could bind to the promoter and enhancer regions of *PIK3C2B* and *SOX4* and induce their expression. Our results suggest that suppression or activation of these genes might be context-dependent[45–49].

To further validate these findings, we examined the *PIK3C2B* and *SOX4* transcriptional levels upon YAP/TAZ siRNA knockdown in different cancer cell lines. YAP and TAZ siRNAs significantly suppress the expression of the known YAP/TAZ target genes (*Cyr61, CTGF* and *ANKRD1*), while *PIK3C2B* and *SOX4* expression levels are significantly upregulated in all these cells (Fig. 4e, f). As expected, both *PIK3C2B* and *SOX4* are elevated in all the cell lines upon MGH-CP1 treatment (Fig. 4g, h). A covalent TEAD inhibitor K975 also induces a significant increase of *PIK3C2B* and *SOX4* transcription levels in H226 cells[22] (Fig. 4h), suggesting that such effects might not be compound specific, but rather common effects of TEAD−YAP blockade[22]. Consistently, overexpression of the constitutive active mutant YAP^S127A or TAZ^S89A in HEK293A cells also lead to suppression of *SOX4* and *PIK3C2B* expression (Fig. 4i). Interestingly, we also found that TEAD1 or TEAD4 could directly bind to the promoter region of *PIK3C2B* through analysis of publicly available ChIP-seq datasets[50] (Supplementary Fig. 6a), suggesting that *PIK3C2B* is a direct target gene of TEADs.

Besides YAP and TAZ, TEADs are known to bind to other co-activators or repressors to regulate gene expression. We previously

showed that TEAD palmitoylation is dispensable to TEAD−VGLL binding[16]. Therefore, we hypothesized that VGLL family of proteins might be involved in the activation of these target genes upon TEAD−YAP blockade. Interestingly, overexpression of VGLL3 could significantly induce *PIK3C2B* and *SOX4* gene expressions, while other VGLL proteins (VGLL1, 2 or 4) have only minor effects (Fig. 4j and Supplementary Fig. 6b–c). Consistently, silencing of VGLL3 reduces *PIK3C2B* and *SOX4* gene expressions (Fig. 4l and Supplementary Fig. 4d). More importantly, siRNA-mediated VGLL3 knockdown could abolish MGH-CP1-induced *PIK3C2B* and *SOX4* expressions (Fig. 4k). Taken together, TEAD−YAP blockade might subject TEAD to bind to other co-factors, and lead to alternative VGLL3-mediated transcriptional activation of *PIK3C2B* and *SOX4*.

## TEAD inhibition activates AKT through PIK3C2B and SOX4

*PIK3C2B*, which encodes one of the members of class II PI3K family, activates downstream AKT pathway, leading to cell survival. It has been reported that SOX4 transcriptionally activates PI3K-AKT pathway[51]. Consistently, amplification of SOX4 in human breast cancers has been shown to promote PI3K-AKT signaling[52], and SOX4 was identified as a critical activator of PI3K-AKT to enable oncogenic survival signals in leukemia[53]. We hypothesized that TEAD inhibition leads to VGLL3-mediated *PIK3C2B* and *SOX4* induction, resulting in AKT activation, promotion of cell survival, and compromising the effects of TEAD−YAP blockade in cancer cells. We analyzed Thr308 and Ser473 phosphorylation of AKT in different cancer cells. Consistently, MGH-CP1 induces strong AKT Thr308 and Ser473 phosphorylation in a time-dependent manner (Fig. 5a). In addition, we found that several reported TEAD−YAP/TAZ inhibitors, including celastrol, TEAD347 and CIL56 could also activate p-AKT[20,54,55] (Fig. 5b). Furthermore, a pan-PI3K inhibitor (wortmannin) could block AKT activation induced by MGH-CP1 in DLD1 and HCT116 cells (Fig. 5c), confirming that activation of PI3K (possibly through PIK3C2B) is involved in TEAD inhibitor-induced AKT activation. Furthermore, SOX4 knockdown also strongly inhibits MGH-CP1-induced AKT phosphorylation (Fig. 5d). These findings suggest that TEAD blockade could activate PI3K-AKT pathway, partly through transcriptional activation of *PIK3C2B* and *SOX4*. In addition, we have carried out immunohistochemistry (IHC) studies of p-AKT (T308 and S473) from mouse liver tissue with MGH-CP1 treatment in vivo. Both wild type and *Lats1/2* deleted mouse liver show low p-AKT signal. Treatment of MGH-CP1 does not induce p-AKT signal in mouse liver tissues, suggesting that the activation of PIK3C2B/SOX4/AKT might be tissue-specific or cancer cell line specific (Supplementary Fig. 7a).

To further understand the role of PIK3C2B/SOX4-AKT activation in the resistance to TEAD inhibitors, we overexpressed PIK3C2B or SOX4 in Huh7, HCT116 and DLD1 cells. Expression of SOX4 or PIK2C2B leads to MGH-CP1 resistance, with SOX4 showing strong effects in Huh7 cells, while PIK2C2B is more effective in colon cancer cell lines (HCT116 and DLD1) (Fig. 5e, f and Supplementary Fig. 6e). Furthermore, knockdown both PIK3C2B and SOX4 could significantly increase the MGH-CP1 sensitivity in Huh7, HCT116 and DLD1 cells (Fig. 5g−h and Supplementary Fig. 6f). Taken together, *PIK3C2B* and *SOX4* induction might confer resistance to TEAD inhibitors in cancer cell lines.

## Synergistic effects of TEAD and AKT inhibitors in cancer cells

As AKT activation is a conserved feedback signaling event induced by TEAD−YAP blockade, we sought to determine whether combination of MGH-CP1 with an AKT inhibitor would have synergistic effects in blocking YAP-dependent cancer cell growth. We performed a drug combination matrix analysis across 5 doses of an AKT inhibitor (ipatasertib) and 9 doses of MGH-CP1. We observed strong synergy in all of the YAP-dependent cancer cell lines tested, including DLD1, HCT116, H226, H1299, Huh7 and HUTU80 (Fig. 6a). In the 3D growth model, we tested drug combination of MGH-CP1 or MGH-CP12 with ipatasertib in

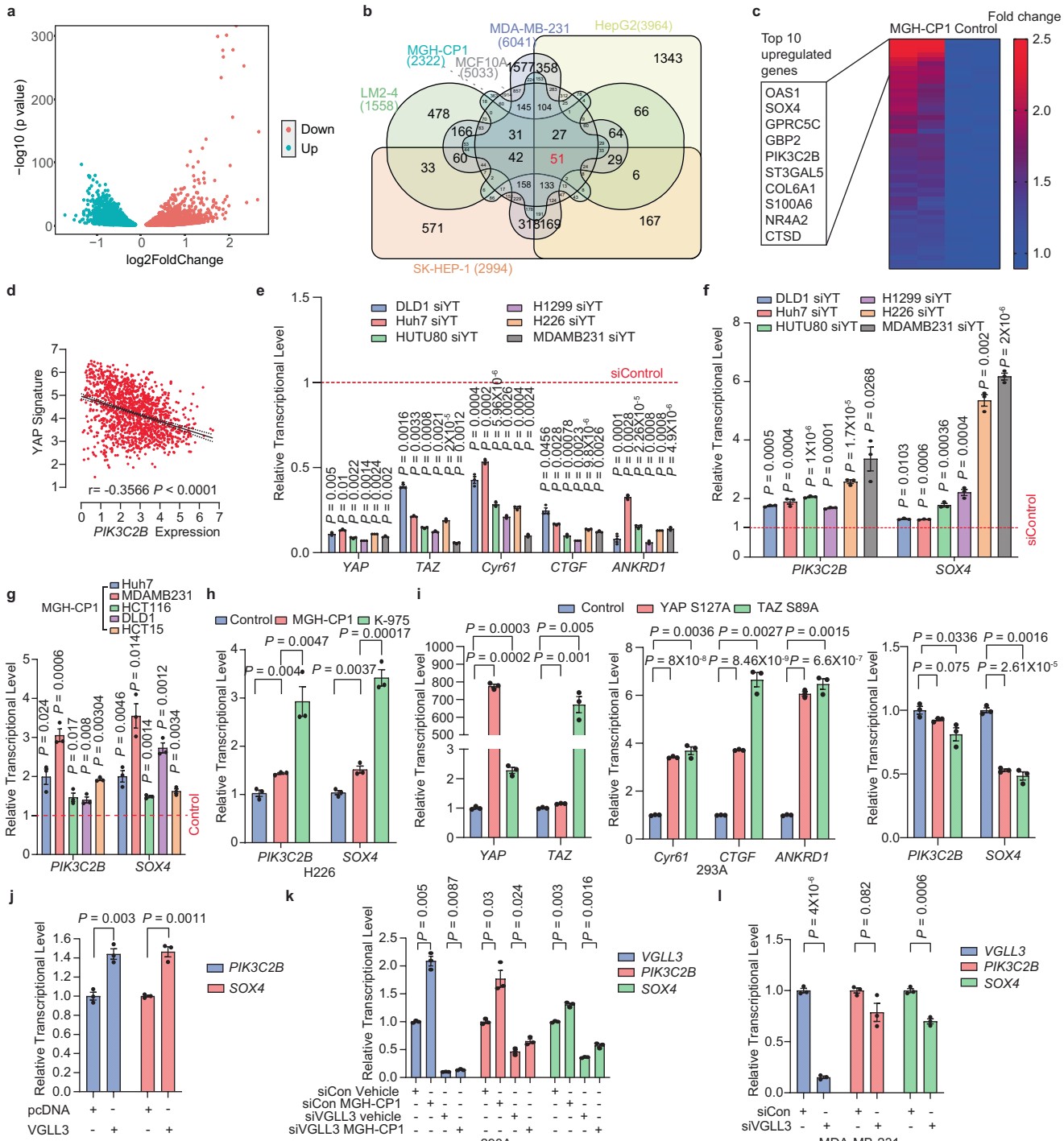

**Fig. 4 | TEAD–YAP/TAZ blockade promotes transcriptional activation of *PIK3C2B* and *SOX4* mediated by VGLL3. a** Volcano plot of RNA-seq results of MDA-MB-231 cells treated with MGH-CP1. **b** Venn diagram of significantly upregulated genes treated with MGH-CP1 and siYAP/TAZ in various cancer cell lines. **c** Heatmap of 51 genes commonly upregulated with TEAD–YAP/TAZ blockade. Top 10 genes with the highest fold change were shown in the box. **d** Pearson correlation of *PIK3C2B* transcriptional level against YAP signature in tumor cells. Transcriptional level of *YAP, TAZ, Cyr61, CTGF, ANKRD1* (**e**), *PIK3C2B* and *SOX4* (**f**) were examined in DLD1, Huh7, HUTU80, H1299, H226 and MDA-MB-231 cells with siYAP/TAZ knockdown. All the gene expression levels of siYAP/TAZ treated samples were normalized to siControl (dashline). (*n* = 3 biological repeats). siYT, siYAP/TAZ. **g** *PIK3C2B* and *SOX4* transcriptional levels in Huh7, MDA-MB-231, HCT116, DLD1 and HCT15 treated with MGH-CP1 treatment were assessed by qPCR (*n* = 3 biological

repeats). **h** H226 cells were treated with MGH-CP1 and K-975, *PIK3C2B* and *SOX4* expression levels were shown (*n* = 3 biological repeats). **i** HEK293A cells were overexpressed with YAP or TAZ, transcriptional levels of *YAP, TAZ, Cyr61, CTGF, ANKRD1, PIK3C2B* and *SOX4* were assessed (*n* = 3 biological repeats). **j** *PIK3C2B* and *SOX4* transcriptional levels in HEK293A cells with overexpression of human VGLL3 (*n* = 3 biological repeats). **k** HEK293A cells were treated with Control siRNA or VGLL3 siRNA in the presence of vehicle control of MGH-CP1, *VGLL3, PIK3C2B* and *SOX4* expression levels were shown (*n* = 3 biological repeats). **l** VGLL3 was silenced with siRNA in MDA-MB-231, and *VGLL3, PIK3C2B* and SOX4 transcriptional levels were determined (*n* = 3 biological repeats). Data are represented as mean ± S.E.M. *P* values were determined using two-tailed *t*-tests. Source data are provided as a Source Data file.

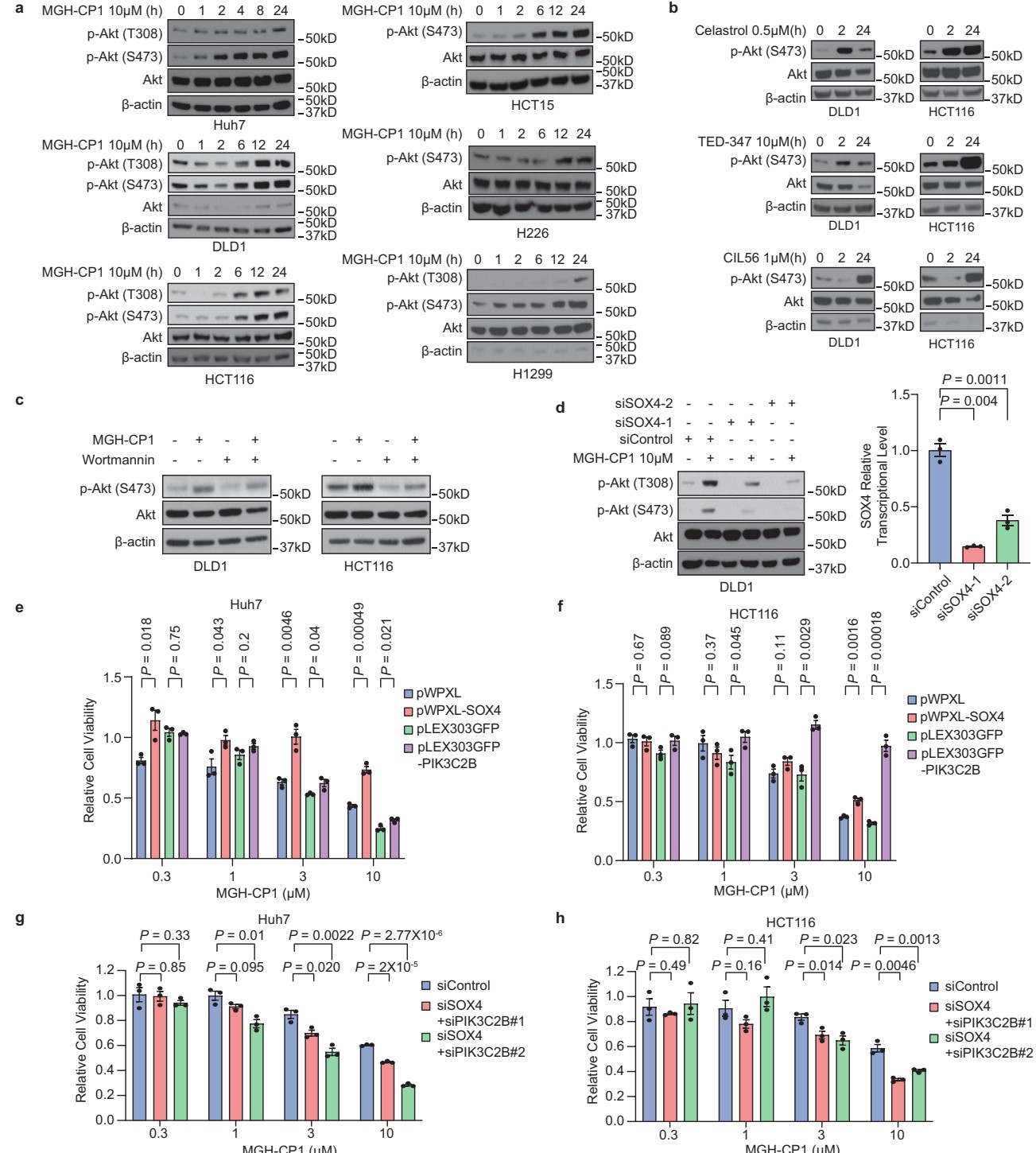

**Fig. 5 | TEAD–YAP blockade activates AKT activity through PIK3C2B and SOX4.**
**a** Immunoblot of p-AKT (T308), p-AKT (S473) and total AKT in Huh7, DLD1, HCT116, HCT15, H226 and H1299 cells treated with MGH-CP1 at indicated time point. **b** Immunoblot of p-AKT (S473) and total AKT in DLD1 and HCT116 cells treated with TEAD-YAP inhibitors, including celastrol, TEAD347 and CIL56 at indicated time point. **c** DLD1 and HCT116 cells were treated with either of MGH-CP1, pan-PI3K inhibitor Wortmannin, or combination. p-AKT (S473) and total AKT levels were determined by immunoblot. **a–c** Representative images were chosen from *n* = 3 biological repeats. **d** DLD1 cells were transfected with sets of SOX2 siRNAs, followed

by MGH-CP1 treatment for 24 h, p-AKT (T308), p-AKT (S473) and total AKT were examined by immunoblot. SOX4 expression levels were evaluated by qPCR after SOX4 siRNA knockdown (*n* = 3 biological repeats). Huh7 (**e**) and HCT116 (**f**) cells were overexpressed with SOX4 or PIK3C2B in the presence of MGH-CP1. Cell viability was shown at different concentrations (*n* = 3 biological repeats). Huh7 (**g**) and HCT116 (**h**) cells were treated with control siRNA and SOX4/PIK3C2B siRNA. Cell viability with treatment of different concentration of MGH-CP1was determined (*n* = 3 biological repeats). Data are represented as mean ± S.E.M. *P* values were determined using two-tailed *t*-tests. Source data are provided as a Source Data file.

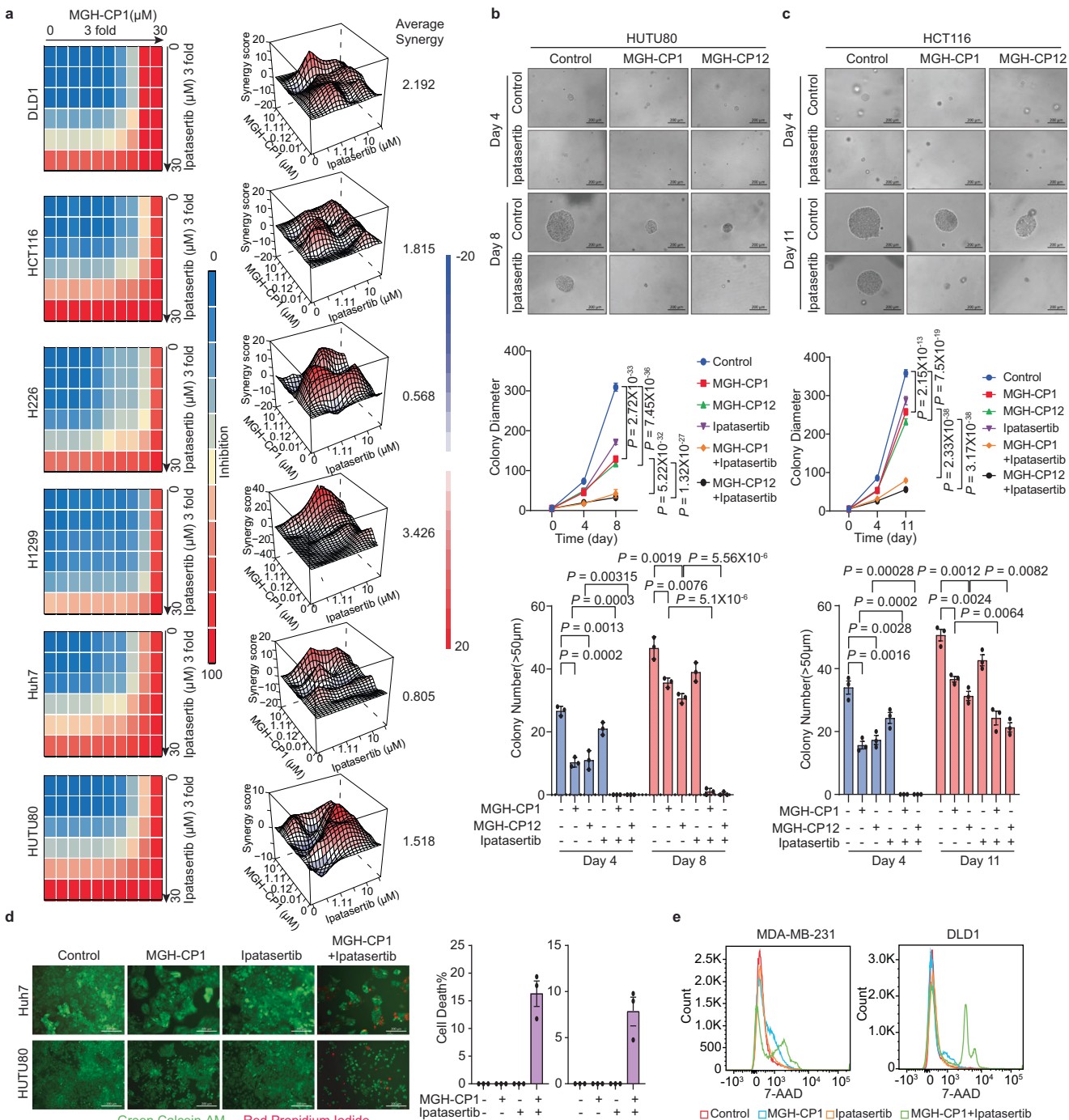

**Fig. 6 | Synergistic effects of TEAD and AKT inhibitors combination in cancer cells. a** Drug combination experiments using MGH-CP1 and AKT inhibitor ipatasertib, and Heatmaps show color-coding as percentage of cell viability normalized to untreated controls. Heatmaps of Bliss score for MGH-CP1 and ipatasertib combination were shown. **b** Representative images in 3D tumor cultures of HUTU80 cells treated with MGH-CP1, MGH-CP12, ipatasertib or combination. Colony diameters were measured to assess the tumor growth with inhibitors (n = 51, 53, 59, 33, 44, 75 colonies for control, CP1, CP12, ipatasertib, CP1 with ipatasertib, and CP12 with ipatasertib, respectively at day 4, n = 104, 94, 75, 97, 74, 77 colonies for the same sample order as above at day 8). Colony numbers were determined (n = 3 biological repeats). Scale bar, 200 μm. **c** Representative images in 3D tumor cultures of HCT116 cells treated with MGH-CP1, MGH-CP12, ipatasertib or

combination. Colony diameters were measured to assess the tumor growth with inhibitors (n = 52, 58, 66, 45, 67, 69 colonies at day 4, n = 68, 79, 85, 58, 66, 84 colonies at day 11, for the same sample order as in **b**. Colony numbers were determined (n = 3 biological repeats). Scale bar, 200 μm. **d** Representative images of fluorescent staining with Calcein-AM and Propidium Iodide in Huh7 and HUTU80 cells treated with MGH-CP1, ipatasertib and combination. Cell death percentages were determined by dead cell/total cells (n = 3 biological repeats). Scale bar, 200 μm. **e** 7-AAD exclusion assay showing the histogram of 7-AAD staining in MDA-MB-231, DLD1 cells treated with MGH-CP1, ipatasertib and combination (representative histograms were chosen from n = 3 biological repeats). Data are represented as mean ± S.E.M. P values were determined using two-tailed t-tests. Source data are provided as a Source Data file.

HUTU80 and HCT116 cells. Although MGH-CP1, MGH-CP12 or ipatasertib alone show some effects in blocking tumor spheres, combination of MGH-CP1 or CP12 with ipatasertib show strong synergy, and significantly reduce tumor spheres sizes and numbers (Fig. 6b, c). To evaluate whether combination treatment could enhance cell killing effects, we stained the live cells with calcein-AM and dead cells with propidium iodide. Combination treatment led to significantly increased cell death compared to single agents at Day 4 in Huh7 and HUTU80 cells (Fig. 6d). Cell killing effects were also confirmed by flow cytometry assay measuring 7-AAD positive cells in MDA-MB-231 and DLD1 cells (Fig. 6e). Taken together, inhibition of the feedback upregulation of PI3K-AKT pathway could synergize with TEAD inhibitor, leading to cell killing effects.

## Discussion

YAP/TAZ hyperactivation is frequently observed in human malignancies[7,56]. Cancer Dependence Map analysis using shRNAs or CRISPR/Cas9 knockout has shown that YAP is involved in a broad range of cancer types, and loss of YAP inhibits the growth of many cancer cell lines. It is of critical need for a validated pharmacological tool to cross-analyze the results obtained from genetic screens. As we demonstrated here, when cross-analyzing the YAP-dependency and MGH-CP1 sensitivity, we found strong and statistically significant correlations between the two datasets, suggesting that MGH-CP1 indeed could recapitulate the effects of genetic deletion or silencing of YAP in a panel of cancer cell lines. However, there are cell lines which are sensitive to YAP shRNAs, but not to MGH-CP1. Follow-up studies would provide information as to whether the discrepancy is due to the off-target effects of either compound or the shRNAs. Besides YAP, TAZ was also implicated as a potent oncogenic factor. Our data showed that MGH-CP1 and its analogues could block both YAP and TAZ-mediated transcription activation. Such effects are not represented in genetic studies using shRNAs targeting YAP or TAZ alone. As there are 4 TEADs (TEAD1-4) in mammals, it has been challenging to genetically disrupt all TEADs. Such pan-TEAD inhibitors could offer unique chemical tools compared to traditional genetic tools to dissect the functions of TEADs in development and cancers. Therefore, close analysis of both YAP and TAZ dependency map, as well as TEAD1-4 dependency in cancer cell lines, will provide more useful information regarding pan-TEAD or subtype specific inhibitors in cancer dependency validation. It has also been noted that MGH-CP1 and other TEAD inhibitors usually only suppress YAP-dependent gene expression levels, such as *CTGF* and *Cyr61*, to 30–50%. Although annotated as YAP-target genes, they are also regulated by other factors, such as Smad3, AP1 and VGLL family of proteins, which might contribute to their high basal expression levels. Therefore, it is important to analyze more broad transcriptome responses using Gene Set Enrichment Analysis, rather than spot checking several target genes in evaluating the effects of compounds.

The transcriptional activity of TEADs is traditionally thought to be regulated through co-activator binding. Besides YAP and TAZ, VGLL1 was shown to bind to TEAD, and ectopic expression of VGLL1 induces target genes expression distinct from TEAD-YAP/TAZ's transcriptional output, enhancing anchorage-independent cell proliferation[57]. TEAD4 directly interacts with transcription factor 4 (TCF4) through its TEA domain to facilitate transactivation of TCF4 and mediate expression of Wnt target genes[58]. The activator protein-1 (AP-1) has been shown to directly interact with TEADs[29,31]. Other binding partners, including poly-ADP ribose polymerase 1 (PARP1), have also been reported[59]. With accumulation of evidence of YAP/TAZ−independent TEADs' functions in regulating gene expression, it is critical to understand how TEADs' association with different factors contributes to different functions in normal development and tumorigenesis. Previously, we found that TEAD palmitoylation is dispensable to TEAD-VGLL4 binding. In addition, MGH-CP1 inhibits TEAD−YAP binding, but not TEAD−VGLL4 association[24]. Here, we also showed that TEAD−YAP blockade could lead to TEAD−VGLL3 mediated transcriptional activation of important oncogenic factors. Therefore, autopalmitoylation of TEAD might regulate TEAD's binding partner selectivity, and MGH-CP1 and its analogues would be important tools to understand how TEAD palmitoylation could regulate its binding to different co-factors and fine-tunes the transcriptional output of Hippo pathway and the crosstalk between Hippo and other important cell signaling pathways.

We also noticed that not all TEAD palmitoylation inhibitors could functionally inhibit TEAD−YAP transcription activities. Some compounds, such as MGH-CP28, do not inhibit TEAD−YAP interaction, albeit potently blocking TEAD2 palmitoylation. Similarly, some reported compounds which inhibit TEAD palmitoylation function as "agonists" to promote TEAD−YAP transcription activities[23]. Therefore, binding to the lipid-binding site of TEAD might have different functional consequences. In our structural studies using TEAD2 YAP binding domain (YBD), we did not observe that compounds could induce significant structural changes. Therefore, the manner by which palmitate or inhibitor binding allosterically alter TEAD functions is still elusive. Further structural studies, including determination of full-length TEAD structures, might offer new insights into the mechanisms.

Recently, several strategies targeting TEAD−YAP complex have been explored, including targeting the protein-protein interaction of TEAD−YAP or TEAD transcriptional activities (verteporfin, VGLL4 peptide, Celastrol)[54,60,61] and TEAD palmitoylation inhibitors (fluofenamic acid, MGH-CP1, TED347, Compound 2, K975, VT compounds etc.)[18,20,22,24–26], these agents provide useful tools to elucidate the functions and utilities of TEAD−YAP blockade. Our and other's results showed that TEAD−YAP blockade could lead to slowdown of tumor growth in vitro and in vivo. However, no significant cell killing was observed. These results are consistent with our observation that TEAD−YAP blockade only leads to transient cell static state rather than cell killing, in most of YAP/TAZ-dependent cancer cell lines. Such lack of cell death effects might be through activation of feedback signals that promote cell survival. Through comprehensive analyses of publicly available transcriptomic datasets with YAP/TAZ knockdown/knockout in multiple cell types[28,41–44], we found that YAP/TAZ inactivation induces expression of a subset of genes, including *PIK3C2B* and *SOX4*, leading to AKT activation and cell survival. Phosphatidylinositol-4-phosphate 3-kinase C2 domain-containing beta polypeptide (PIK3C2B) is an enzyme encoded by *PIK3C2B* gene, and belongs to the class II PI3K family. PIK3C2B and the downstream AKT activation have been manifested in regulating multiple cellular processes involved in cell proliferation, oncogenic transformation, cell survival, cell migration, and intracellular protein trafficking. Previously, *PIK3C2B* was reported to be a YAP-inducible gene through TEAD modulation in MCF10A cells[49]. In addition, *SOX4* has also been linked to Hippo/YAP regulating, involved in promoting stemness, proliferation and inhibiting apoptosis[45,46,48]. SOX4 is a critical transcription factor that regulates cell fate through modulating developmental pathways, including PI3K, Wnt and TGFβ signaling[51]. Interestingly, we found that VGLL3 is a major mediator of transcriptional induction of *PIK3C2B* and *SOX4*, suggesting that TEAD−VGLL3 activation might bypass TEAD−YAP blockade (Supplementary Fig. 8). The VGLL family of proteins are known to bind to TEADs, functioning as either transcription activator or repressors, and share many target genes with YAP/TAZ. However, the functions of VGLLs in cancer and other diseases are not well defined. Previously, VGLL3 has been shown to enhance breast cancer cell proliferation and drive systemic autoimmunity[62,63]. Recently, VGLL3 has been shown to bind to TEAD and suppress *ESR1* gene expression[64]. Here we show that VGLL3 functions as a transcription co-activator to induce the expression of YAP-suppressed genes, suggesting complexed and context dependent functions of VGLL family of proteins.

Recently, several small molecule inhibitors of TEADs have been reported, and three compounds are currently in Phase I clinical trials. These compounds showed significant anti-tumor effects in NF2-mutant mesothelioma cells, while having limited effects on other YAP/TAZ-dependent cancer cell lines. It would be interesting to explore whether combination of AKT inhibitors could enhance their therapeutic activities in a broader range of cancers.

## Methods

### MGH-CP1 and analogue compounds synthesis

Schemes for Synthesis route 1 to 5 are shown in the Supplementary Fig. 9.

Synthesis route 1

4-((1$S$,3$R$,5$S$)-adamantan-1-yl)−2-bromoaniline: To a solution of 4-((1$S$,3$R$,5$S$)-adamantan-1-yl)aniline (200 mg, 0.88 mmol, 1 equiv) in DMF (3 mL) was added NBS (164.4 mg, 0.92 mmol, 1.05 equiv) in DMF (1 mL). The reaction mixture was stirred for 15 mins. The mixture was then diluted with ethyl acetate and extracted with water. The organic layer was washed with brine, dried over Na$_2$SO$_4$. and concentrated on a rotary evaporator. The crude was used without further purification.

Synthesis route 2

To a solution of aniline in anhydrous DCM was added K$_2$CO$_3$ followed by addition of acyl chloride dropwise on ice. The reaction mixture was stirred at rt. The progress of reaction was monitored by TLC. Then, the reaction was quenched by saturated NaHCO$_3$ and diluted with DCM. The combined organic layers were washed with brine, dried over Na$_2$SO$_4$. and concentrated on a rotary evaporator. The crude was used without further purification.

Synthesis route 3

To a solution of thiol in DMSO was added K$_2$CO$_3$ (1.5 equiv). After stirred at rt for 5 min, the suspension was added substituted chloride (1.2 equiv). The reaction mixture was stirred at rt overnight. The progress of reaction was monitored by TLC. The reaction was quenched by water and diluted with ethyl acetate. The combined organic layers were washed with brine, dried over Na$_2$SO$_4$. and concentrated on a rotary evaporator. The residue was purified by silica gel flash column chromatography to give the desired product.

2-((4$H$−1,2,4-triazol-3-yl)thio)-$N$-(4-((1$S$,3$R$,5$S$)-adamantan-1-yl) phenyl)acetamide (**MGH-CP1**). MGH-CP1 (69%) was obtained from 4$H$−1,2,4-triazole-3-thiol and $N$-(4-((1$S$,3$R$,5$S$)-adamantan-1-yl)phenyl)−2-chloroacetamide as white solid. $^1$H NMR (500 MHz, DMSO-$d_6$) $\delta$ 10.17 (s, 1H), 8.41 (s, 1H), 7.45 (d, $J$ = 8.5 Hz, 2H), 7.25 (d, $J$ = 8.5 Hz, 2H), 3.99 (s, 2H), 2.01 (s, 3H), 1.79 (d, $J$ = 2.5 Hz, 6H), 1.73–1.64 (m, 6H). $^{13}$C NMR (126 MHz, DMSO-$d_6$) $\delta$ 166.51, 146.69, 136.56, 125.34, 119.43, 43.00, 36.50, 35.70, 28.64.

2-((4$H$−1,2,4-triazol-3-yl)thio)-$N$-(4-(tert-butyl)phenyl)acetamide (**MGH-CP2**). **MGH-CP2** (54%) was obtained from 4$H$−1,2,4-triazole-3-thiol and $N$-(4-(tert-butyl)phenyl)−2-chloroacetamide as white solid. $^1$H NMR (500 MHz, CDCl$_3$) $\delta$ 9.62 (br s, 1H), 8.15 (s, 1H), 7.40 (d, $J$ = 8.5 Hz, 2H), 7.29 (d, $J$ = 9.0 Hz, 2H), 3.87 (s, 2H), 1.26 (s, 9H). $^{13}$C NMR (126 MHz, CDCl$_3$) $\delta$ 167.85, 148.00, 134.86, 125.86, 120.17, 36.43, 34.40, 31.30, 31.21.

2-((4$H$−1,2,4-triazol-3-yl)thio)-$N$-([1,1'-biphenyl]−4-yl)acetamide (**MGH-CP9**). **MGH-CP9** (77%) was obtained from 4$H$−1,2,4-triazole-3-thiol and $N$-([1,1'-biphenyl]−4-yl)−2-chloroacetamide as white solid. $^1$H NMR (500 MHz, DMSO-$d_6$) $\delta$ 10.37 (s, 1H), 8.47 (br s, 1H), 7.73–7.59 (m, 6H), 7.44 (t, $J$ = 7.5 Hz, 2H), 7.32 (t, $J$ = 7.5 Hz, 1H), 4.08 (s, 2H). $^{13}$C NMR (126 MHz, DMSO-$d_6$) $\delta$ 166.32, 139.66, 138.43, 135.09, 128.93, 127.08, 127.00, 126.28, 119.46, 36.34.

2-((4$H$−1,2,4-triazol-3-yl)thio)-$N$-(4-((1$S$,3$R$,5$S$)-adamantan-1-yl) −2-bromophenyl)acetamide (**MGH-CP12**). **MGH-CP12** (90%) was obtained from 4$H$−1,2,4-triazole-3-thiol and $N$-(4-((1$S$,3$R$,5$S$)-adamantan-1-yl)−2-bromophenyl)−2-chloroacetamide as pale brown solid. $^1$H NMR (500 MHz, CDCl$_3$) $\delta$ 9.27 (s, 1H), 8.17 (s, 1H), 8.08 (d, $J$ = 8.5 Hz, 1H), 7.45 (d, $J$ = 2.5 Hz, 1H), 7.29–7.24 (m, 1H), 3.96 (s, 2H), 2.07 (s, 3H),

1.82 (d, $J$ = 2.5 Hz, 6H), 1.79–1.67 (m, 6H). $^{13}$C NMR (126 MHz, CDCl$_3$) $\delta$ 167.74, 157.55, 149.91, 145.94, 132.70, 129.18, 124.84, 122.63, 114.76, 42.98, 36.56, 36.29, 36.05, 28.77.

3-((4$H$−1,2,4-triazol-3-yl)thio)-$N$-(4-((1$S$,3$R$,5$S$)-adamantan-1-yl) phenyl)propanamide (**MGH-CP27**). **MGH-CP27** (66%) was obtained from 4-methyl-4$H$−1,2,4-triazole-3-thiol and $N$-(4-((1$S$,3$R$,5$S$)-adamantan-1-yl)phenyl)−3-chloropropanamide as white solid. $^1$H NMR (500 MHz, DMSO-$d_6$) $\delta$ 9.88 (s, 1H), 8.43 (br s, 1H), 7.49 (d, $J$ = 8.0 Hz, 2H), 7.26 (d, $J$ = 8.5 Hz, 2H), 3.33 (t, $J$ = 7.0 Hz, 2H), 2.75 (t, $J$ = 7.0 Hz, 2H), 2.04 (s, 3H), 1.82 (d, $J$ = 2.5 Hz, 6H), 1.76–1.67 (m, 6H). $^{13}$C NMR (126 MHz, DMSO-$d_6$) $\delta$ 168.97, 145.78, 136.57, 124.84, 118.91, 42.68, 36.43, 36.20, 35.33, 28.33, 27.08.

4-((4H-1,2,4-triazol-3-yl)thio)-$N$-(4-((1$S$,3$R$,5$S$)-adamantan-1-yl) phenyl)butanamide (**MGH-CP34**). **MGH-CP34** (84%) was obtained from 4$H$−1,2,4-triazole-3-thiol and $N$-(4-((1$S$,3$R$,5$S$)-adamantan-1-yl) phenyl)−4-chlorobutanamide as white solid. $^1$H NMR (500 MHz, DMSO-$d_6$) $\delta$ 9.85 (s, 1H), 8.39 (br s, 1H), 7.49 (d, $J$ = 8.0 Hz, 2H), 7.25 (d, $J$ = 8.0 Hz, 2H), 3.13 (t, $J$ = 7.5 Hz, 2H), 2.42 (t, $J$ = 7.5 Hz, 2H), 2.03 (s, 3H), 1.99–1.91 (m, 2H), 1.82 (d, $J$ = 2.5 Hz, 6H), 1.77–1.67 (m, 6H). $^{13}$C NMR (126 MHz, DMSO-$d_6$) $\delta$ 170.20, 170.11, 145.63, 136.73, 136.63, 124.78, 118.94, 42.68, 36.20, 35.31, 34.97, 31.02, 28.32, 25.34.

$N$-(4-((1$S$,3$R$,5$S$)-adamantan-1-yl)phenyl)−2-((4-methyl-4$H$−1,2,4-triazol-3-yl)thio)acetamide (**MGH-CP8**). **MGH-CP8** (77%) was obtained from 4-methyl-4$H$−1,2,4-triazole-3-thiol and $N$-(4-((1$S$,3$R$,5$S$)-adamantan-1-yl)phenyl)−2-chloroacetamide as white solid. $^1$H NMR (500 MHz, DMSO-$d_6$) $\delta$ 10.20 (s, 1H), 8.55 (s, 1H), 7.45 (d, $J$ = 8.5 Hz, 2H), 7.28 (d, $J$ = 9.0 Hz, 2H), 3.02 (s, 2H), 3.60 (s, 3H), 2.04 (s, 3H), 1.82 (d, $J$ = 3.0 Hz, 6H), 1.77–1.68 (m, 6H). $^{13}$C NMR (126 MHz, DMSO-$d_6$) $\delta$ 165.56, 148.71, 146.23, 136.23, 124.99, 118.98, 42.64, 37.74, 36.18, 35.37, 30.84, 28.31.

2-((1,3,4-thiadiazol-2-yl)thio)-$N$-(4-((1$S$,3$R$,5$S$)-adamantan-1-yl) phenyl)acetamide (**MGH-CP28**). **MGH-CP28** (82%) was obtained from 1,3,4-thiadiazole-2-thiol and $N$-(4-((1$S$,3$R$,5$S$)-adamantan-1-yl)phenyl) −2-chloroacetamide as white solid. $^1$H NMR (500 MHz, DMSO-$d_6$) $\delta$ 10.31 (s, 1H), 9.51 (s, 1H), 7.49 (d, $J$ = 7.0 Hz, 2H), 7.29 (d, $J$ = 7.0 Hz, 2H), 4.30 (s, 2H), 2.04 (s, 3H), 1.83 (m, 6H), 1.76–1.67 (m, 6H). $^{13}$C NMR (126 MHz, DMSO-$d_6$) $\delta$ 165.10, 164.95, 154.33, 146.36, 136.21, 125.05, 119.06, 42.67, 38.34, 36.20, 35.41, 28.33.

Synthesis route 4

2-((4$H$−1,2,4-triazol-3-yl)sulfinyl)-$N$-(4-((1$S$,3$R$,5$S$)-adamantan-1-yl)phenyl)acetamide (**MGH-CP25-1**). To a solution of 2-((4$H$−1,2,4-triazol-3-yl)thio)-$N$-(4-((1$S$,3$R$,5$S$)-adamantan-1-yl)phenyl)acetamide (100 mg, 0.27 mmol, 1 equiv) in THF (2 mL) was added m-CPBA (70.2 mg, 0.32 mmol, 1.2 equiv, 77%) on ice. After stirred at 0 °C for 40 min, the reaction mixture was quenched with saturated Na$_2$S$_2$O$_3$ solution. The mixture was then diluted with ethyl acetate and extracted. The combined organic layers were washed with water and brine, dried over Na$_2$SO$_4$. and concentrated on a rotary evaporator. The residue was purified by silica gel flash column chromatography to give **5** (32.3 mg, 31%) as white solid. $^1$H NMR (500 MHz, DMSO-$d_6$) $\delta$ 10.41 (s, 1H), 8.86 (s, 1H), 7.49 (d, $J$ = 8.5 Hz, 2H), 7.29 (d, $J$ = 9.0 Hz, 2H), 4.53 (d, $J$ = 14.0 Hz, 1H), 4.27 (d, $J$ = 13.5 Hz, 1H), 2.04 (s, 3H), 1.83 (d, $J$ = 3.0 Hz, 6H), 1.77–1.66 (m, 6H). $^{13}$C NMR (126 MHz, DMSO-$d_6$) $\delta$ 164.10, 162.62, 146.54, 146.04, 135.98, 125.03, 119.06, 59.39, 42.65, 36.18, 35.41, 28.32.

Synthesis route 5

2-((4$H$−1,2,4-triazol-3-yl)sulfonyl)-$N$-(4-((1$S$,3$R$,5$S$)-adamantan-1-yl)phenyl)acetamide (**MGH-CP25**). To a solution of 2-((4$H$−1,2,4-triazol-3-yl)thio)-$N$-(4-((1$S$,3$R$,5$S$)-adamantan-1-yl)phenyl)acetamide (100 mg, 0.27 mmol, 1 equiv) in DCM/THF (1.5/1 mL) was added m-CPBA (187.3 mg, 0.81 mmol, 3 equiv, 77%) on ice. After stirred at rt overnight, the reaction mixture was quenched with saturated Na$_2$S$_2$O$_3$ solution. The mixture was then diluted with ethyl acetate and extracted. The combined organic layers were washed with water and brine, dried over Na$_2$SO$_4$. and concentrated on a rotary evaporator. The residue was purified by silica gel flash column chromatography to give

**MGH-CP25** (69%) as white solid. $^1$H NMR (500 MHz, DMSO-$d_6$) $\delta$ 10.34 (s, 1H), 8.90 (s, 1H), 7.43 (d, $J$ = 9.0 Hz, 2H), 7.29 (d, $J$ = 8.5 Hz, 2H), 4.55 (s, 2H), 2.03 (s, 3H), 1.82 (d, $J$ = 3.0 Hz, 6H), 1.76–1.66 (m, 6H). $^{13}$C NMR (126 MHz, DMSO-$d_6$) $\delta$ 160.70, 158.45, 146.75, 146.15, 135.83, 125.05, 119.06, 60.83, 42.63, 36.17, 35.42, 28.31.

All reagents used in the methods section are listed in Supplementary Table 2.

## MGH-CP1 Pharmacokinetics assay
(MGH-CP1 has poor metabolic stability in mouse liver microsome assays with less than 20% remining after 60 min incubation. In intravenous pharmacokinetic studies (1 mg/kg), the half-life is estimated as ~0.3 h.)

## Molecular modeling methods
Molecules were prepared with LigPrep default protocol in Schrodinger Suite (2021-1 release). Co-crystal structure of TEAD2 protein (pdbcode: 6CDY) was prepared and then used to generate docking grid. Constraint Glide docking with enhanced sampling was applied where hydrogen bonding with GLN410 was required. Top 5 poses for each molecule were examined visually and one pose was chosen manually to represent the putative binding mode.

## In vitro palmitoylation of recombinant TEAD protein
Recombinant His$_6$-TEAD2 (500 ng) protein was pretreated with MGH-CP compounds at indicated doses for 0.5 h followed by incubation with the 1 µM of alkyne palmitoyl-CoA (Cayman Chemical) for 0.5 h in 50 mM MES buffer (pH 6.4). Click reaction was performed as described previously[65]. Briefly, CuSO$_4$/TBTA/TCEP/Biotin-Azide master mix was added into 50 µL protein/palmitoyl-CoA reaction buffer, making the final concentration CuSO$_4$ 100 µM, TBTA 10 µM, TCEP 100 µM and Biotin-Azide 10 µM. Samples were incubated for 1 h at room temperature, followed by SDS-PAGE analysis. Biotinylated TEAD protein was detected by streptavidin-HRP. Band intensities obtained from streptavidin blots were quantified using ImageJ (NIH).

## Cell culture
MDAMB231(Cat# HTB-26), NCI-H226 (Cat#CRL-5826) were from ATCC. HEK293A, NCI-H1299, SK-HEP1, A375, SK-Mel2, SK-Mel28, SK-Mel3, Huh7, SNU449, SNU398, DLD1, HCT116, HUTU80, MCF10A were obtained from the cell line repository at MGH cancer center cell line repository. OMM1.3, Mel202 were the gift from Xu Chen' lab in University of California San Francisco, 92.1 and OCM1 were the gift from Kun-Liang Guan's lab in University of California San Diego. HEK293A, Huh7, SNU398, SNU449, MDA-MB-231, A375, SK-MEL-2, SK-MEL-28, SK-MEL-31, HCT116, DLD1, HUTU80 cell lines were cultured in Dulbecco's modified Eagles media (DMEM) (Life Technologies) supplemented with 10% FBS (FBS) (Thermo/Hyclone, Waltham, MA) and 100 units/mL penicillin/100 µg/mL streptomycin. OMM1.3, Mel202, OCM1, NCI-H1299 and NCI-H226 cell lines were cultured in RPMI 1640 supplemented with 10% FBS and 100 units/mL penicillin/100 µg/mL streptomycin. None of the cell lines used in this paper are listed in the database of commonly misidentified cell lines maintained by ICLAC. All cell lines are free of mycoplasma contamination. All cell lines used in this study were authenticated by identification of short tandem repeat (STR) markers. All the cells from the MGH cancer center cell line repository were authenticated by SNPs and STR analysis to exclude cross-contaminated or synonymous lines.

## Labeling, click reactions and streptavidin pull-down
HEK293A cells were labeled with DMSO or probe (Alkynyl Palmitic acid) in medium with 10% fatty acid free FBS overnight. The cells were lysed in lysis buffer (50 mM TEA-HCl, pH 7.4, 150 mM NaCl, 1% Triton X-100, 0.2% SDS, cOmplete EDTA-free protease inhibitors) followed by

Click reaction with biotin-Azide[65]. Proteins were precipitated with 9 volumes of cold methanol overnight at −80 °C, and then recovered by centrifugation at 17,000 × g for 10 min. The precipitates were suspended in suspension buffer (PBS, 0.05% tween-20 and 2% SDS) and diluted with IP buffer (PBS, 0.05% tween-20). Labeled cellular proteins were enriched using streptavidin agarose (Life Technologies) at room temperature with rotation for 4 h. Protein-bound streptavidin agarose beads were washed three times with PBST (0.05% tween-20), and bound proteins were eluted with elution buffer (10 mM EDTA pH 8.2 and 95% formamide) at 95 °C for 10 min. Samples were processed with 6X SDS-PAGE sample buffer and proteins were resolved by SDS-PAGE. TEAD1 or TEAD4 in these samples were detected using anti-myc-tag and anti-pan-TEAD antibodies.

## Western blotting
Cells were lysed with RIPA or indicated lysis buffer supplemented with protease inhibitors (Roche) and phosphatase inhibitors (Roche). Lysates were denatured by heating for 5 min at 95 °C and loaded onto 4−12% Bis-Tris polyacrylamide gel. MES running buffer (Invitrogen) was used for the SDS-PAGE. The proteins were subsequently transferred to polyvinylidene fluoride (PVDF) membranes (Millipore). The membranes were blocked and incubated with primary antibodies and secondary HRP-conjugated antibodies, and developed by exposure to film. Antibody and dilutions used in the studies: Myc-tag (EMD Millipore, Cat#MABE282, clone 9E10, 1:1,000), Myc-Tag (Cell Signaling, Cat#2278 S, clone 71D10, 1:1,000), Flag-tag (Cell Signaling Cat#2368 S,Binds to same epitope as Sigma's Anti-FLAG® M2 Antibody, 1:1,000), Flag-tag (Sigma, Cat#F1804, clone M2, 1:1,000), HA-tag (Cell Signaling Cat#3724 S, clone C29F4, 1:1,000), His-Tag (Invitrogen Cat#MA1-21315, clone HIS.H8, 1:1,000), β-actin (ABCAM Cat#ab6276, clone AC-15, 1:5,000), Streptavidin-HRP (Life Technologies Cat#S911, 1:5,000), YAP/TAZ (Cell Signaling Cat#8418 S, clone D24E4, 1:1,000), p-Akt (S473) (Cell Signaling Cat#4060 S, clone D9E, 1:1,000), p-Akt(T308) (Cell Signaling Cat#9275 S, 1:1,000), Akt (Cell Signaling Cat#2920 S, clone 40D4, 1:1,000), Anti-Rabbit HRP (Cell Signaling Cat#7074 S, 1:5,000), Anti-Mouse HRP (Cell Signaling Cat#7076 S, 1:5,000). The antibody dilution information can be found in Supplementary Information (Supplementary Table-Reagent and Primer list). All uncropped gel images are available in Supplementary Figs. 10−15.

## Transfection
Plasmids were transfected with PEI (1 µg/µL). Briefly, cells were seeded a day before. DNA was diluted in the serum-free DMEM and mixed with PEI (DNA: PEI ratio = 1:2). After incubation for 20 min at room temperature, mixture was added directy into the wells. Expression level was determined 48 h later.

siRNAs were transfected with Lipofectamine RNAiMAX following manufacturer's protocol. Briefly, 25 pmol RNAi duplex (10uM stock) was mixed with 5 uL Lipofectamine RNAiMAX in Opti-MEM reduced Serum Medium without serum. RNAi duplex-Lipofectamin RNAiMAX complexes were added into cells in 6 well plate format. Knockdown efficiency was examined 36−48 h later. siRNA sequences used were: GACAUCUUCUGGUCAGAGA dTdT (YAP), ACGUUGACUUAGGAACUUU dTdT (TAZ). MISSION® siRNA Universal Negative Control #1 (Sigma Cat#SIC001) was used as siRNA control.

## Co-immunoprecipitation
HEK-293A cells were transfected with the indicated constructs. The compounds were administrated at indicated concentrations on the following day. After 24 h, cells were lysed by sonication with lysis buffer (50 mM Tris-HCl pH 7.5, 10% Glycerol, 1% NP-40, 300 mM NaCl, 150 mM KCl, 5 mM EDTA, phosphatase inhibitor cocktail, cOmplete EDTA-free protease inhibitors cocktail). Extracts were diluted with 50 mM Tris-HCl pH 7.5, 10% Glycerol, 1% NP-40, 5 mM EDTA. Flag-YAP or Myc-TEAD1/4 was immunoprecipitated with anti-FLAG M2 magnetic

beads or anti-c-myc antibody, respectively, overnight with rotation at 4 °C. TEAD1/4 was captured using Protein A/G magnetic resins. Protein-bound resins were washed three times with lysis buffer and processed with SDS-PAGE sample buffer. Blots were probed with anti-myc, anti-FLAG antibodies.

## Luciferase assay

Gal4-UAS-Luc and the expression vectors for YAP, Gal4-TEAD1, Gal4-TEAD2[66] as well as Renilla luciferase constructs were transfected into HEK293A cells and, 24 h post-transfection, cells were treated with MGH-CP1 overnight and processed using the homemade dual-glow luciferase assay protocol[67]. Briefly, cells were lysed in buffer with 25 mM Tris-phosphate pH 7.8, 2 mM DTT, 2 mM 1,2-diaminocyclo-hexane-N,N,Ń ,Ń -tetra acetic acid, 10% glycerol and 1% Triton® X-100. Extracts were processed in 96 well white-walled plate followed by adding substrate of firefly luciferase (25 mM Tris-phosphate pH 7.8, 1 mM luciferin, 3 mM ATP, 15 mM MgCl, 0.2 mM coenzyme A and 1 M DTT). Renilla luciferase activity was measured by substrate buffer (0.01 mM h-CTZ and 0.06 mM PTC124, 45 mM Na2EDTA, 30 mM Pyrophosphate tetrabasic and 1.425 M NaCl). Luminescence of Firefly and Renilla luciferase activities were quantified using PerkinElmer EnVision plate reader.

## Tumor migration assay

Cells were seeded into 24-well Corning transwell and incubated with control or MGH-CP1 at indicated concentrations (Corning Cat# CLS3464). Twenty-four hours later, migrating cells were stained with Crystal violet and quantitated.

The Oris Cell Migration Assay uses a 96-well plate with "stopper" barriers that create a central cell-free Detection Zone for cell migration experiments (PLATYPUS CAT# CMA1.101). Removing the stoppers allows the cells to migrate into the Detection Zone at the center of each well. Calcein-AM was applied to stain the viable cells.

## in vitro Tumor sphere formation

Stem like cells were enriched from MCF10A, KNS62 and Huh7 by culturing in serum-free DMEM-F12 medium (Life Technologies) containing 50 µg/ml insulin (Sigma-Aldrich), B-27 Supplement (Life Technologies), 20 µg/ml EGF (STEMCELL), 20 µg/ml basic FGF (STEMCELL) and in 4 ug/mL heparin (STEMCELL) ultra-low attachment flasks (Corning) to support growth of undifferentiated oncospheres. Once tumor spheres formed, they were dissociated by trypsinization into single cells that would be reseeded at low density for tumor sphere formation again. After multiple passages, the tumor sphere formation ability was measured in the presence of indicated compounds. The first passage of tumor sphere formation was labeled with "Primary", and after multiple passages, the tumor sphere was labeled with "Secondary". Images were captured using Zeiss microscope. Spheres were counted and plotted as shown.

## Generation of stable cell lines

shRNAs of YAP and TAZ were cloned into Tet-pLKO-puro (Addgene 21915). A 10 cm dish of 80% confluent HEK293FT cells was transfected with 10 µg of the transfer plasmid, 5 µg shRNA, 3 µg psPAX2, 2 µg VSV-G and 20 µL of PEI transfection reagent in Optim-MEM without serum. Media was changed after overnight incubation. After 48 h, viral supernatants were filtered through a 0.45 µm low protein binding membrane (Millipore) and used immediately supplied with polybrene. Transduction was performed in Huh7 cells, followed by selection with 2 µg/mL puromycin for 1 weeks. The sequences of shYAP and shTAZ used were:

shYAP: CCGGGCGATGAATCAGCCTCTGAATCTCGAGATTCAGA GGCTGATTCATCGCTTTTT

shTAZ: CCGGGCCACCAAGCTAGATAAAGAACTCGAGTTCTTTA TCTAGCTTGGTGGCTTTTT

## Tumor cell 3D Culture

SeaPrep agarose was used for cell culture. A 6%(w/v) agarose solution in DPBS was prepared and the final concentration diluted in media for cell culture was 1%. A mixture of cells and agarose was prepared from cell and agarose stocks pre-warmed to 37 °C, and dispensed into ultra-low-attachment 96-well plates (Corning Cat#3474). The plates were incubated at 4 °C for 15–30 min to allow gelling to occur. After gelling of the agarose, the cultures were incubated at 5% CO2 and 37 °C in a humidified incubator. Tumor cell growth was monitored under microscope. More than 50 random colonies in each group were recorded at indicated time point by Zeiss Axio photo observer microscope. The diameter of each tumor colony was measured through ZEN 3.2 blue edition software.

## Quantitative RT-PCR

Total RNA was extracted using the RNeasy mini kit (Qiagen, Cat#74104) or trizol according to manufacturer's instruction. cDNA was synthesized from 2 µg of total RNA using a high-capacity cDNA reverse transcription kit (Life Technologies Cat# 4368814) with random primers, according to the manufacturer's protocol. Gene expression was quantified using PowerUp SYB Green Master Mix kit (Life Technologies A25777) in the Roche lightcycler 480 System and normalized to GAPDH/β-actin. The primers used in this study are provided in Supplementary Table 3.

## Cell cycle

Cells were harvested and washed in PBS, followed by Fixing in cold 70% ethanol for at least 30 min at 4 °C. Cells were washed by Spinning at 2000 rpm and resuspended in 1xPBS three times. 50 µl of 100 µg/ml RNase was added to remove RNA. Cells were added with 425 µl of cell staining buffer (1% BSA in PBS) and 25 µl of Propidium Iodide Solution for flow cytometry analysis to quantify DNA content. Cell cycle analysis was performed by FlowJo. Gating strategy of FACS can be found in the Supplementary Fig. 16.

## Protein purification

The cDNA encoding human TEAD2 (residues 217–447, TEAD2$_{217-447}$) was cloned into a modified pET29 vector (EMD Biosciences) that included a C-terminal His$_6$-tag. The construct was verified by DNA sequencing. The pET29-TEAD2$_{217-447}$ plasmid was transformed into the E. coli strain BL21(DE3)-T1$^R$ cells (Sigma) for protein expression. His$_6$-tagged TEAD2$_{217-447}$ was purified with Ni$^{2+}$-NTA agarose resin (Qiagen) and then purified by anion exchange chromatography with a resource-Q column followed by size exclusion chromatography with a Superdex 75 column (GE Healthcare). Purified TEAD2$_{217-447}$ was concentrated to 4 mg/ml in a buffer containing 20 mM Tris (pH 8.0), 100 mM NaCl, 2 mM MgCl$_2$, 1 mM TCEP and 5% glycerol.

## Drug screen across large cell line collection

High-throughput drug screening and sensitivity modeling (curve fitting and IC$_{50}$ estimation) was performed essentially as described previously[36]. Cells were grown in RPMI or DMEM/F12 medium supplemented with 5% FBS and penicillin/streptomycin and maintained at 37 °C in a humidified atmosphere at 5% CO2. Cell lines were propagated in these two media in order to minimize the potential effect of varying the media on sensitivity to therapeutic compounds in our assay, and to facilitate high-throughput screening. To exclude cross-contaminated or synonymous lines, a panel of 92 SNPs was profiled for each cell line (Sequenom, San Diego, CA) and a pair-wise comparison score calculated. In addition, short tandem repeat (STR) analysis (AmpFlSTR Identifiler, Applied Biosystems, Carlsbad, CA) was performed and matched to an existing STR profile generated by the providing repository. Briefly, cells were seeded in 384 well plates at variable density to ensure optimal proliferation during the assay. Drugs were added to the cells the day after seeding for adherent cell

lines and the day of seeding for suspension cell lines. For tumor subtypes containing both adherent and suspension cells, all lines where drugged the same day (small cell lung cancer cell lines for example were all drugged the day after seeding). A series of nine doses was used using a 2-fold dilution factor for a total concentration range of 256 fold. Viability was determined using resazurin after 5 days of drug exposure. 360 genomically-unique cell lines were screened.

## Analysis of YAP dependency and MGH-CP1 growth inhibition sensitivity of human cancer cell lines

Cancer Dependency Map was completed by Eli and Edythe L. Broad Institute of MIT and Harvard recently (https://depmap.org)[37,38,68]. It is a systematic effort aimed at identifying and cataloging gene essentiality across hundreds of genomic characterized cancer cell lines. The project uses genome-scale RNAi and CRISPR-Cas9 genetic perturbation reagents to silence or knock out individual genes and identify those genes that affect cell survival. ATLANTIS, a nonlinear regression modeling method, was developed to find molecular markers that are predictive of DEMETER dependency scores, which is for finding and characterizing predictive biomarker-dependency models using the R package "party" to build an ensemble of conditional inference[69]. The cancer cell lines show more dependency with lower DEMETER dependency scores. We took advantage of these public data[37] and correlated with MGH-CP1 $IC_{50}$. The correlation was completed by GraphPad prism.

## Mouse intestinal organoid culture

Intestinal organoids were generated from the small intestinal crypts isolated from two-month-old mice using the method previously described[70]. Briefly, mouse intestine was dissected and intestinal epithelium was separated by ice-cold 5 mM EDTA-PBS. Following vigorously shaking, intestinal crypts were collected by centrifuge at 4 degree. The intestinal crypts were then resuspended and cultured in the Geltrex® Matrix (Thermo Fisher) in the presence of R-Spondin, Noggin and EGF (Proteintech), and split 4 times before MGH-CP1 treatment. The complete medium and the inhibitor were changed every 2 days.

## AAV infection and liver enlargement model

The animals use protocols were reviewed and approved by The University of Massachusetts Medical School Institutional Animal Care and Use Committee. Adeno-Associated Virus expressing Cre recombinase (AAV9-Cre) was purchased from Penn Vector Core (University of Pennsylvania Perelman School of Medicine) and delivered into $Lats1^{f/f};Lats2^{f/f}$ mice through intraperitoneal injection. One day after, MGH-CP1 (75 mg/kg) or DMSO was administrated via daily intraperitoneal injection for 21 days in $Lats1^{f/f};Lats2^{f/f}$ mice with or without AAV-Cre injection.

## Xenograft tumor model

All xenograft tumor studies were conducted in accordance with NIH animal use guidelines and a protocol approved by MGH IACUC (Protocol# 2013N000065). Mouse Room Condition: Light cycle: 12 light/12 dark cycle is used; Temperature:18–23 °C; Humidity: 40–60%. The mice were obtained from Gnotobiotic Mouse Cox 7 Core (Massachusetts General Hospital). For xenograft tumor establishment, 5 million Huh7 cells were inoculated bilaterally into the posterior back region of the 6-8 weeks female SCID (NOD -$Prkdc^{-/-}$) mice. For 92.1 xenograft tumors, 5 million of 92.1 cells mixed with Matrigel were inoculated bilaterally into the posterior back region of the 6-8 weeks female SCID (NOD -$Prkdc^{-/-}$) mice. For MDA-MB-231 cells, 2 million cells were inoculated bilaterally into the posterior back region of the 6-8 weeks female NOD -$Prkdc^{-/-}$Il2rg$^{-/-}$ mice. For Huh7 initiation experiment 1, MGH-CP1 started to treat the mice one day post-inoculation for two weeks at 25 mg/kg and 50 mg/kg. Once treatment had been suspended, tumors were measured for three weeks. For Huh7 initiation

experiment 2, tumor cells were treated for 24 h at 10 μM MGH-CP1 or DMSO control before inoculation. Five million Huh7 cells were inoculated bilaterally into the posterior back region of the 6-8 weeks female SCID (NOD -$Prkdc^{-/-}$) mice. Tumors were monitored and measured for tumor volume. For MDA-MB-231 initiation experiment 1, MGH-CP1 started to treat the mice one day post-inoculation for two weeks at 75 mg/kg. Once treatment had been suspended, tumors were measured for three weeks. For MDA-MB-231 initiation experiment 2, tumor cells were treated for 48 h at 10 μM MGH-CP1, MGH-CP12 or DMSO control before inoculation.

For treatment experiment, tumors were administered with MGH-CP1 daily i.p. at 50 mg/kg once tumor had been established. All the tumor volumes were measured using caliper. Tumor volume was calculated using the formula: Tumor volume $(mm^3) = d^2 x\ D/2$ where d and D are the shortest and longest diameter in mm, respectively.

The mice would be euthanized by $CO_2$ when tumors reach 1.5 cm at any direction according to guideline of MGH IACUC. The maximal tumor size in this study does not exceed the tumor size permitted.

## Immunohistochemistry and immunofluorescence staining

For immunohistochemistry (IHC), sections were deparaffinized and rehydrated before undergoing heat-induced antigen retrieval in 10 mM sodium citrate buffer (pH 6.0) for 30 min. Slides were blocked for endogenous peroxidase for 20 min, then blocked for 1 h in 5% BSA, 1% goat serum, 0.1% Tween-20 buffer in PBS, and incubated overnight at 4 °C in primary antibody diluted in blocking buffer or SignalStain® Antibody Diluent (Cell Signaling). Slides were incubated in biotinylated secondary antibodies for 1 h at room temperature and signal was detected using the Vectastain Elite ABC kit (Vector Laboratories). Ki67 (Cell Signaling Cat#9027 S) was used at dilution 1:500. For immunofluorescence (IF), cells or tissue sections were fixed by 4% paraformaldehyde for 5 min, blocked for 1 h and incubated overnight at 4 °C in primary antibody diluted in blocking buffer. Slides were then incubated for 1 h at room temperature in Alexa Fluor-488 secondary antibody (Invitrogen R37118) at 1:500 dilution in blocking buffer and mounted using mounting media with DAPI (EMS).

## RNA-seq

MDA-MB231 cells were treated with MGH-CP1 at 10 mM for 24 h. Total RNA was isolated with RNeasy Mini Kit. The integrity of isolated RNA was analyzed using Agilent 2100 Bioanalyzer, and the RNA-seq libraries were made by Novogene. All libraries have at least 50 million reads sequenced (150 bp paired-end). The raw RNA-seq data of YAP/TAZ siRNA knockdown in MDA-MB-231 cells were obtained from the previously published report[28]. The correlation between gene expression changes in MGH-CP1 treated and YAP/TAZ siRNA knockdown MDA-MB231 cells was performed using Pearson correlation analysis. The P values of genes changes in control groups compared with MGH-CP1 or YAP/TAZ siRNA were determined by Student's $t$ test. Plots of correlation between fold change was generated by ggplot2 package in R. Principle component analysis (PCA) was determined and plotted by M3C package in R. Gene Set enrichment analysis was performed using GSEA software from Broad Institute and UC San Diego. YAP/TAZ-TEAD molecular signatures database was used according to the previous published report[29].

## Drug combination

The drug combination experiments were preformed using a drug combination matrix across 5 doses of Ipatasertib (3-fold dilution) and 9 doses of MGH-CP1 (3-fold dilution) in different tumor cell lines. Cell viability were determined at day 5 after the drugs administration by MTT. Drug synergy score was calculated followed Bliss rule. Plot was generated by Synergyfinder package in R.

## Statistics and reproducibility

No statistical method was used to predetermine sample size. The experiments were not randomized. For biochemical experiments we performed the experiments at least three independent times. Experiments for which we showed representative images were performed successfully at least 3 independent times. No samples or animals were excluded from the analysis. The investigators were not blinded to allocation during experiments and outcome assessment. All data are shown as mean ± standard error of the mean (S.E.M.). All $P$ values were determined using two-tailed $t$-tests and statistical significance was set at $P = 0.05$. Time courses were analyzed by repeated measurements (mixed model) ANOVA with Bonferroni post-tests. The variance was similar between groups that we compared. Graphs were generated by GraphPad prism 8.0.

## Data availability

The RNA-seq dataset generated for the current study is available in the GEO repository with access code GSE177052. The public RNA-seq dataset used for analysis during the current study is available in GSE102407 (PMID: 30224758)[28], GSE49384 (PMID: 24581491)[42], GSE54617 (PMID: 26389641)[43], GSE56445 (PMID: 24648515)[41],GSE59229 (PMID: 25796446)[44]. The TEAD1/4 ChIP data are from UCSC ENCODE data (https://www.encodeproject.org/search/?searchTerm=TEAD)[50]. Co-crystal structure of TEAD2 protein is available at PROTEIN DATA BANK (pdbcode: 6CDY)[24]. Dependency scores are available from Depmap portal (https://depmap.org/portal/)[37]. All uncropped gel images are provided in the Supplementary Information File. Source data are provided with this paper. The remaining data are available within the Article, Supplementary Information or Source Data file. Source data are provided with this paper.

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

## Acknowledgements

This work was supported by grants from National Institutes of Health R01CA219814 (X.W.) and R01CA238270 (to X.W. and J.M.), R01DK127180 and R01DK127207 to J.M., C.H.B. was supported by a grant from the Wellcome Trust (102696). L.H. is partly supported by a postdoctoral fellowship from the Antidote foundation for cure of cancer. Z.T. is supported by Melanoma Research Alliance (MRA) postdoctoral fellowship.

## Author contributions

X.W. designed the experiments, conceived and supervised the studies. Y.S. designed and performed the biochemistry, cell biology, xenograft tumors experiments and analyzed the profiling data with the assistance of H.E. L.H. and G.J. performed chemicals synthesis and SAR study. Z.T. performed biochemistry and cancer cell biology studies. A.S, Q.L., and J.C. performed compound testing in intestinal organoids and Lats1/2 knockout animal models both in vitro and in vivo with input from J.M. and Y.T.I. J.M. contributed to experimental design and analysis of Lats1/2 knockout mice experiments. C.H.B., P.G. and R.K.E. contributed the cancer cell line profiling. J.C. performed compound docking studies. Y.S., L.H, Z.T., J.M. and X.W. wrote the manuscript with input from all co-authors.

## Competing interests

X.W. has a financial interest in Tasca Therapeutics, which is developing small molecule modulators of TEAD palmitoylation and transcription factors. Dr. Wu's interests were reviewed and are managed by Mass General Hospital, and Mass General Brigham in accordance with their conflict of interest policies. The remaining authors declare no competing interests.
