## [Peer Review File · Nature Communications]

Reviewers' Comments:

Reviewer #1:

Remarks to the Author:

This MS by Sun et al., provides a further characterization of a compound, MGH-CP1, that the authors isolated in a previous publication (PMID: 32259481) as an inhibitor of the TEAD transcription factors, by blocking their palmitoylation and their interaction with the oncogenic transcription cofactors YAP and TAZ.

The results show that MGH-CP1 partially inhibits YAP and TAZ activities in cancer cells and affects various malignant properties in culture, and tumor growth and initiation in vivo.

The authors then suggest that the partial and temporary effects of MGH-CP1 treatment is linked to an increase in the activity of the PI3K/Akt pathway they consistently observed after inhibiting YAP/TAZ activity by several methods. The authors propose that this is a consequence of upregulation of PI3K and SOX4 after YAP/TAZ inactivation. In line with these observations, the authors also found that treatments with the Akt inhibitor Ipatasertib is synergistic with MGH-CP1 in inducing cancer cell death, at least in vitro.

Data presented in this MS seem mostly incremental as compared to the original publication, to the point that some of the figures of this new MS are taken directly from PMID: 32259481 with just minimal changes (for example, compare Fig. 2a of this MS with Figure S4L of the previous publication).

The only part that displays a significant element of novelty is the upregulation of the PI3K/Akt pathway after YAP/TAZ inhibition. However, this part provides only minimal information and the proposed mechanism is poorly substantiated. Indeed, YAP/TAZ depletion, or MGH-CP1 treatment, induces only a very minor upregulation, generally less than two-fold, of PIK3C2B and SOX4 mRNA expression (see Fig. 4d), that is hardly compatible with the upregulation of the Akt activity reported in Fig. 5.

Additional points (listed below) also dampened the enthusiasm for this story:

A) The authors state that (lines 201-203) "MGH-CP1 shares significant overlap with YAP/TAZ siRNA in modulating gene expression, and suggest that MGH-CP1 is a specific TEAD-YAP/TAZ inhibitor.". In support of such a statement, the authors compared the transcriptome gene expression data from MDA-MB-231 breast cancer cells depleted of YAP/TAZ or treated with MGH-CP1, using Principal Component Analysis (Fig. 2a) or linear correlation (Fig. 2b). Although suggestive, these analyses are per se insufficient to support the claim of a significant overlap of the effects of MGH-CP1 with those of YAP/TAZ depletion. Instead, the authors have to provide a direct comparison between the genes that are significantly downregulated and upregulated by the two treatments, and use Fisher's test or similar statistical assays to determine whether they indeed share a significant overlap. This is particularly important to reinforce the notion that MGH-CP1 is a specific TEAD-YAP/TAZ inhibitor. This is also important to verify if MGH-CP1-treatment induces transcriptional effects that are of the same magnitude of those induced by YAP/TAZ depletion.

B) To evaluate the effects of TEAD inhibitors on tumor cell self-renewal, the authors measured the effects of MGH-CP1 on formation of spheroid structures by cells seeded on low-attachment plates. This assay, however, is not sufficient to make a claim on the self-renewal abilities of these cells, as pointed out in several publications (see for example PMID 21549325). The most compelling assay to make such a claim is measuring the ability of these spheres to be propagated for multiple passages after dissociation and reseeding of single cells at low density, to allow formation of clonal structures instead of aggregates.

C) It is disturbing that cell lines from different types of cancer are used for different types of experiments, perhaps suggesting that the reported effects of MGH-CP1 were not consistently the same in all tested cell lines. For instance, the effects of MGH-CP1 on transcription and on in vivo tumor formation were tested in the breast cancer cell line MDA-MB-231 and in the hepatocellular carcinoma cell line Huh7, respectively. At least, the authors should have shown that MGH-CP1 is effective against tumor initiation of MDA-MB-231 cells.

Reviewer #3:

Remarks to the Author:

The authors provide a logical series of methodologies to demonstrate the major findings of their work, namely that inhibition of TEAD palmitoylation does suppress cell growth in cancer cells (transiently), due to activation of the PIK3C2B/SOX4/Akt pathway. The data are sound and well presented, and provide a basis of action for cancer treatment with a combined inhibition of TEAD palmitoylation and concomitant inhibition of the AKT pathway.

Two minor points should be addressed:

1. Even though the mechanisms are different, the authors should also provide some information as to whether in normal (non-cancer) liver tissue (liver with double Lats1 and Lats 2 KO), suppression of the Hippo pathway and consistent elevation of non-phosphorylated Yap also results in activation of the PIK3C2B/SOX4/Akt pathway in the Lats1/2 Ko liver.

2. In Fig.1g: The transcription of Cyr61, CTGF, ANKRD1 is indeed suppressed by MGH-CP, but not abolished. What other transcription related factors for these proteins maintain a ~30%-50% level of continuing transcription not dependent on Yap-TEAD?

Reviewer #4:

Remarks to the Author:

In this study, Sun et al examine the consequences of inhibiting autopalmitylation of the TEAD transcription factor on cancer cell growth in vitro and in vivo. Starting with their previously identified small molecule inhibitor, MGH-CP1, the authors performed SAR and identified a related compound, MGH-CP12 with 10-fold higher potency in inhibiting TEAD autopalmitylation. Both compounds inhibited TEAD-YAP interaction and transcriptional activities, as well as tumor sphere formation. The authors continued in vivo analyses with MGH-CP1, and showed that drug treatment slowed tumor initiation and tumor growth in a YAP-dependent manner. However, cell death was not observed, leading the authors to explore changes in gene expression that correlated with YAP/TAZ knockdown in cancer cell lines treated with MGH-CP1. RNA seq analysis uncovered a mechanism of feedback regulation whereby TEAD inhibition (via YAP/TAZ knockdown) activates PI3K/Akt and SOX4 signaling pathways. The authors conclude that dual inhibition of TEAD plus Akt might be useful.

This is a carefully performed, in depth study that provides new insights into the mechanism whereby the TEAD-YAP pathway regulates cancer cell growth. In all drug treatment studies, there is always the concern of potential off-target effects. Ideally one would test this in cells in which TEAD has been knocked out, but this is not feasible due to the presence of multiple forms of the TEAD proteins (TEAD1-4) (as noted in lines 409-410). The authors have done the next best thing and used YAP knockdown cells to establish YAP-dependency for MGH-CP1. The finding of feedback regulation is particularly interesting and opens up an option for combination therapy in YAP-dependent cancers.

Two issues should be addressed in a revised manuscript.

1. One concern is the message that the authors want to convey with regard to potency for MGH-CP1 vs MGH-CP12. IC50 curves presented for inhibition of in vitro and cell-based palmitoylation (Fig 1b, 1d) clearly show that MGH-CP12 is more effective than MGH-CP1 at inhibiting TEAD palmitoylation (10-fold lower IC50). However, in Fig 1e, the authors state that MGH-CP12 "is more potent than MGH-CP1" in blocking TEAD/YAP interactions, but the differences appear minimal in the bar graph. A similar conclusion is made for the tumor sphere assays in Fig 1i, where little difference between MGH-CP1 and CP12 is apparent. No statement is made with regard to compound potency for Figs 1f and 1g, and again, there are minimal differences between the two compounds. Since these assays were performed at high compound concentrations (10 uM), far above the IC50's for both compounds, it is not possible to directly compare potencies. If the authors wish to conclude that MGH-CP12 is the "better" inhibitor, then they need to perform

assays where the response is titrated vs increasing concentrations of compound. They should also explain why all further experimentation was only performed with MGH-CP1 (ie, why did they not use CP12 if its better).

2. There is an apparent lack of correlation between inhibition of TEAD palmitoylation and transcriptional and biological activities by the different CP compounds. For example, MGH-CP27 and 28 are good TEAD palmitoylation inhibitors, but have minimal ability to block the TEAD-YAP interaction (Fig 1e). The authors suggest this might be due to the allosteric nature of the lipid binding site, but do not provide any evidence for how this is impacting the action (or lack thereof) of MGH-CP27 and 28.

Minor issues:

3. Suppl Fig 1d,e: The authors suggest that "inhibition of TEAD palmitoylation has very limited effects on TEAD protein stability" but ref 16 (Noland et al) demonstrated that TEAD palmitoylation stabilizes the proteins. Please explain the apparent discrepancy.

4. The IC50 curve for CP12 should be expanded in more detail (in Supplemental), as plot in Fig 1 does not contain enough data points in the titratable range.

5. The manuscript would benefit from editing for English language usage.

Response to Reviewers' comments

Reviewer #1:

We appreciate the comments and the constructive suggestions from the reviewer. We have performed additional experiments to address all these concerns. We have now gained additional mechanistic insights into Sox4/PI3K/Akt induction by MGH-CP1. In addition, several new *in vivo* experiments were performed as the reviewer suggested. Below please find our point-by-point responses to Reviewer 1's comments.

- 1) *“some of the figures of this new MS are taken directly from PMID: 32259481 with just minimal changes (for example, compare Fig. 2a of this MS with Figure S4L of the previous publication).”*

Response:

The RNA-seq results used in this manuscript were new datasets recreated in MDA-MB-231 cells treated with MGH-CP1 (GSE177052), which are different from the previous one published (PMID: 32259481, GSE140396). The new dataset (GSE177052) has improved overall data quality, and provides additional insights into the mechanism. The data were analyzed by PCA and Pearson correlations with YAP/TAZ siRNA. Per Reviewer 1's suggestion, we performed Fisher's exact test, and showed that MGH-CP1 and siYAP/TAZ indeed share a significant overlap in both upregulated and downregulated gene signatures. The new Fisher's test panel replaced the PCA and Pearson correlations in previous Figure 2a and b (see **new Figure 2a**).

- 2) *“The only part that displays a significant element of novelty is the upregulation of the PI3K/Akt pathway after YAP/TAZ inhibition. However, this part provides only minimal information and the proposed mechanism is poorly substantiated. Indeed, YAP/TAZ depletion, or MGH-CP1 treatment, induces only a very minor upregulation, generally less than two-fold, of PIK3C2B and SOX4 mRNA expression (see Fig. 4d), that is hardly compatible with the upregulation of the Akt activity reported in Fig. 5.”*

Response:

We appreciate the suggestions from Reviewer 1, and performed many new experiments to substantiate the mechanistic insights. As TEADs also bind to VGLL family of proteins, disruption of TEAD-YAP might lead to alternative TEAD-VGLL-mediated transcription. Indeed, we previously showed that palmitoylation is dispensable to TEAD-VGLL binding. To understand the detailed mechanisms, we examined the role of VGLLs (VGLL1-4) in regulation of *PIK3C2B* and *SOX4* expression. Interestingly, overexpression of VGLL3 significantly promotes the expression of *PIK3C2B* and *SOX4* (**new Figure 4j**). Consistently, knockdown of VGLL3 could attenuate the MGH-CP1-induced *PIK3C2B* and *SOX4* expression in multiple cell lines (HEK293, MDA-MB-231 and H226) (**See new Figure 4k and 4l, Supplementary Figure 5d**). These results suggest that VGLL3 functions as a transcriptional co-activator involved in the modulation of *PIK3C2B* and *SOX4* expression, upon MGH-CP1 treatment. Therefore, disrupting TEAD-YAP interaction by MGH-CP1 might lead to alternative TEAD-VGLL3 activation.

In addition, we found that expressing PIK3C2B or SOX4 in cancer cell lines confers resistance to MGH-CP1 (new Figure 5a and b, Supplementary Figure 6e). Consistently, knockdown SOX4 and PIK3C2B together by siRNA could sensitize cancer cell lines to the MGH-CP1 treatment (new Figure 5c and Supplementary Figure 6f). These results suggest that VGLL3-mediated PIK3C2B and SOX4 upregulation might contribute to the resistance to TEAD inhibition.

To confirm the contribution of PIK3C2B and SOX4 upregulation to the induction of phospho-AKT, we inhibited PIK3C2B with wortmannin, or silenced SOX4 with siRNA, and observed that phospho-AKT induction was substantially attenuated. These results indicate that MGH-CP1-mediated PIK3C2B and SOX4 upregulation lead to AKT activation (new Figure 5c and d).

Furthermore, we analyzed the publicly available ChIP-seq data (GSE66081 and UCSC), and found that TEAD binding peaks were shown at PIK3C2B regulatory region, that indicates PIK3C2B is a direct target gene of TEAD, which could be modulated by TEAD-YAP or TEAD-VGLL complex (new Supplementary Figure 5a).

We believe these new results substantially improved our manuscript, and provided additional mechanistic insights into resistance development upon TEAD inhibition.

“Additional points (listed below) also dampened the enthusiasm for this story:

A) The authors state that (lines 201-203) “MGH-CP1 shares significant overlap with YAP/TAZ siRNA in modulating gene expression , and suggest that MGH-CP1 is a specific TEAD-YAP/TAZ inhibitor.”. In support of such a statement, the authors compared the transcriptome gene expression data from MDA-MB-231 breast cancer cells depleted of YAP/TAZ or treated with MGH-CP1, using Principal Component Analysis (Fig. 2a) or linear correlation (Fig. 2b). Although suggestive, these analyses are per se insufficient to support the claim of a significant overlap of the effects of MGH-CP1 with those of YAP/TAZ depletion. Instead, the authors have to provide a direct comparison between the genes that are significantly downregulated and upregulated by the two treatments, and use Fisher’s test or similar statistical assays to determine whether they indeed share a significant overlap. This is particularly important to reinforce the notion that MGH-CP1 is a specific TEAD-YAP/TAZ inhibitor. This is also important to verify if MGH-CP1-treatment induces transcriptional effects that are of the same magnitude of those induced by YAP/TAZ depletion.”

Response:

We sincerely thank reviewer’s suggestions. We performed Fisher’s exact test to determine MGH-CP1 and siYAP/TAZ indeed share a significant overlap. We observed a significant overlap between MGH-CP1 treatment and siYAP/TAZ knockdown. The new Fisher’s test panel replaced the PCA and Pearson correlations in original main Figure 2a and b (new Figure 2a).

B) To evaluate the effects of TEAD inhibitors on tumor cell self-renewal, the authors measured the effects of MGH-CP1 on formation of spheroid structures by cells seeded on low-attachment plates. This assay, however, is not sufficient to make a claim on the self-renewal abilities of these cells, as pointed out in several publications (see for example PMID 21549325). The most compelling assay to make such a claim is measuring the ability of these spheres to be propagated for multiple passages after dissociation and reseeding of single cells at low density, to allow formation of clonal structures instead of aggregates.

Response:

Per the reviewer 1's suggestion, we assessed primary and secondary tumor sphere formation upon compound treatment. We observed that MGH-CP1 and MGH-CP12 inhibit primary tumor sphere formation with IC50s of 720nM and 260nM, respectively. For secondary sphere formation, MGH-CP1 and CP12 are even more potent with IC50s of 230nM and <100nM, respectively (**new Figure 1h and Supplementary Figure 1f**). We believe that these new data sufficiently demonstrated that TEAD inhibition potently block tumor stem-like cells self-renewal.

“C) It is disturbing that cell lines from different types of cancer are used for different types of experiments, perhaps suggesting that the reported effects of MGH-CP1 were not consistently the same in all tested cell lines. For instance, the effects of MGH-CP1 on transcription and on in vivo tumor formation were tested in the breast cancer cell line MDA-MB-231 and in the hepatocellular carcinoma cell line Huh7, respectively. At least, the authors should have shown that MGH-CP1 is effective against tumor initiation of MDA-MB-231 cells.”

Response:

We thank for reviewer's comments and suggestions. We performed multiple new *in vivo* experiments, including tumor initiation experiments with MDA-MB-231 cells. The mice were inoculated with MDA-MB-231 cells, and MGH-CP1 (75mg/kg) were dosed in the following day. Tumor formation and growth were monitored and recorded (**Supplementary Figure 4g-i**). We found that CP1 treatment significantly inhibits MDA-MB-231 tumor initiation. Since we do not have PK information for MGH-CP12, MGH-CP1 was the only compound chosen for *in vivo* experiments.

We also performed the tumor initiation experiment with mice inoculated with MDA-MB-231 cells that were pre-treated with vehicle control, MGH-CP1 or MGH-CP12 at 10 μ M for 48 hours *in vitro*. The tumor formation and growth were measured (**Supplementary Figure 4j-l**). Pre-treatment of the cells with both TEAD inhibitors also significantly reduced the tumor formation *in vivo*, confirming that TEAD inhibition could block tumorigenic activities of MDA-MB-231 cells.

In addition to review 1's suggestion of MDA-MB-231 tumor experiment, we have also included the *in vivo* tumor growth inhibition using YAP-dependent *GNAQ/GNA11* mutant uveal melanoma (UM) cell 92.1. The results showed that MGH-CP1 treatment could significantly inhibit 92.1 UM tumor growth *in vivo*, correlating with YAP-target gene inhibition (**Supplementary Fig. 4c-f**).

We believe that these additional *in vivo* experiments sufficiently addressed Review 1's comments.

Reviewer #3:

We appreciate the comments from Reviewer 3 that “the data are sound and well presented, and provide a basis of action for cancer treatment with a combined inhibition of TEAD palmitoylation and concomitant inhibition of the AKT pathway.”

“Two minor points should be addressed:

1. Even though the mechanisms are different, the authors should also provide some information as to whether in normal (non-cancer) liver tissue (liver with double Lats1 and Lats 2 KO), suppression of the Hippo pathway and consistent elevation of non-phosphorylated Yap also results in activation of the PIK3C2B/SOX4/Akt pathway in the Lats1/2 Ko liver.”

Response:

We have carried out IHC studies of p-AKT (T308 and S473) from mouse liver tissue after MGH-CP1 treatment. Both wild type and Lats1/2 deleted mouse liver show low p-AKT signal. Treatment of CP1 does not induce p-AKT signal in normal mouse tissues (Supplementary Fig. 7). These results suggest that the activation of PIK3C2B/SOX4/AKT might be tissue-specific or cancer cell line specific.

“2. In Fig.1g: The transcription of Cyr61, CTGF, ANKRD1 is indeed suppressed by MGH-CP, but not abolished. What other transcription related factors for these proteins maintain a ~30%-50% level of continuing transcription not dependent on Yap-TEAD?”

Response:

We thank for reviewer's comments and suggestions. Similar to other reported TEAD inhibitors (VT103, K-975 etc.), inhibition of TEADs does not completely suppressed YAP/TAZ-TEAD target genes, such as *Cyr61*, *CTGF* and *ANKRD1*. This phenomenon is observed from many TEAD inhibitors as well^{1, 2, 3, 4, 5}. As reported, other Hippo-independent coactivators such as AP1^{6, 7}, TCF4⁸, SMAD^{9, 10} could regulate transcription of *Cyr61*, *CTGF* and *ANKRD1* that are independent on YAP/TAZ-TEAD association, which might account for the incomplete suppression of *CTGF/Cyr61/ANKRD1* expression. We have added a discussion of these points.

Reviewer #4 (Remarks to the Author):

We thank reviewer's positive comments that "this is a carefully performed, in depth study that provides new insights into the mechanism whereby the TEAD-YAP pathway regulates cancer cell growth", and "the finding of feedback regulation is particularly interesting and opens up an option for combination therapy in YAP-dependent cancers."

"Two issues should be addressed in a revised manuscript.

1. One concern is the message that the authors want to convey with regard to potency for MGH-CP1 vs MGH-CP12. IC50 curves presented for inhibition of in vitro and cell-based palmitoylation (Fig 1b, 1d) clearly show that MGH-CP12 is more effective than MGH-CP1 at inhibiting TEAD palmitoylation (10-fold lower IC50). However, in Fig 1e, the authors state that MGH-CP12 "is more potent than MGH-CP1" in blocking TEAD/YAP interactions, but the differences appear minimal in the bar graph. A similar conclusion is made for the tumor sphere assays in Fig 1i, where little difference between MGH-CP1 and CP12 is apparent. No statement is made with regard to compound potency for Figs 1f and 1g, and again, there are minimal differences between the two compounds. Since these assays were performed at high compound concentrations (10 μ M), far above the IC50's for both compounds, it is not possible to directly compare potencies. If the authors wish to conclude that MGH-CP12 is the "better" inhibitor, then they need to perform assays where the response is titrated vs increasing concentrations of compound. They should also explain why all further experimentation was only performed with MGH-CP1 (ie, why did they not use CP12 if its better)."

Response:

We thank reviewer's suggestion, and performed additional experiments to evaluate CP1 and CP12.

1) In addition to TEAD2 autopalmitylation assay, we performed *in vitro* TEAD4 autopalmitylation assay using MGH-CP12 (New Supplementary Figure 1b). The IC50s of MGH-CP12 in TEAD2 and TEAD4 autopalmitylation assays are 0.182 μ M and 0.851 μ M respectively, compared with MGH-CP1's IC50s of 0.71 μ M (TEAD2) and 0.672 μ M (TEAD4). These results suggest that CP12 is more potent than CP1 in TEAD2 assay, but slightly less potent than CP1 in TEAD4 assay.

2) Endogenous pan-TEAD palmitoylation was evaluated in HEK293A cells with MGH-CP1 and MGH-CP12. We observed that MGH-CP12 is only slightly better to inhibit pan-TEAD autopalmitylation compared to MGH-CP1 (New Supplementary Figure 1c).

3) We performed TEAD2-YAP binding luciferase assay using MGH-CP1 and MGH-CP12. the IC50 values are 1.047 μ M and 0.302 μ M, respectively, which are consistent with TEAD2 autopalmitylation assay results (New Supplementary Figure 1d). However, In the TEAD-binding element reporter gene assay, the IC50 values of CP1 and CP12 are 1.687 μ M and 0.907 μ M, respectively (New Supplementary Figure 1e), suggesting that both compounds have similar potency in blocking overall TEADs transcription activities, with CP12 only slightly better.

Taken together, these results indicate that MGH-CP12 is more potent to TEAD2 than TEAD4. As we previously used CP1 in cancer cell line profiling and RNA-seq analysis, to be consistent, we chose to use CP1 for most of the mechanistic studies and in vivo work.

“2. There is an apparent lack of correlation between inhibition of TEAD palmitoylation and transcriptional and biological activities by the different CP compounds. For example, MGH-CP27 and 28 are good TEAD palmitoylation inhibitors, but have minimal ability to block the TEAD-YAP interaction (Fig 1e). The authors suggest this might be due to the allosteric nature of the lipid binding site, but do not provide any evidence for how this is impacting the action (or lack thereof) of MGH-CP27 and 28.”

Response:

We appreciate the comments of the reviewer. It has been noted that compounds bind to the lipid-binding pocket could allosterically inhibit or activate TEAD-YAP complex. The detailed mechanisms are still elusive, and possibly require structural information of the full-length TEAD proteins. It is possible that CP27 and CP28 bind to the pocket, but do not induce sufficient conformational change to inhibit TEAD-YAP binding. In addition, work from Genentech and Hong lab at A*STAR (NUS) showed that the correlation of palmitoylation inhibition to functional inhibition/activation of TEAD-YAP is not clear^{2, 11}. We have added a discussion in the manuscript to address these questions. We and others in the field are actively working on it to understand the detailed mechanisms.

“Minor issues:

3. Suppl Fig 1d,e: The authors suggest that “inhibition of TEAD palmitoylation has very limited effects on TEAD protein stability” but ref 16 (Noland et al) demonstrated that TEAD palmitoylation stabilizes the proteins. Please explain the apparent discrepancy.”

Response:

We thank reviewer’s comments. In our assay, mutation of palmitoylation site or treatment of palmitoylation inhibitor did not affect TEAD protein stability. These results were also confirmed by other groups using a radiolabeled pulse-chase protein half-life experiment². They found that there is no appreciable difference in TEAD half-life in the presence of palmitoylation inhibitor. Similarly, other TEAD inhibitors (VT103 and others, including Genentech’s Compound 2)^{2, 5} do not significantly decrease the stability of TEAD proteins. Collectively, inhibition of TEAD palmitoylation has limited effects on TEAD protein stability. We have added a discussion of these results.

“4. The IC50 curve for CP12 should be expanded in more detail (in Supplemental), as plot in Fig 1 does not contain enough data points in the titratable range.”

Response:

We thank reviewer’s comments and suggestions.

We performed TEAD2 autopalmitylation assay with more doses of MGH-CP12, as well as TEAD4 (**New Supplementary Figure 1b**). The IC50s of MGH-CP12 in TEAD2 and TEAD4

autopalmitoylation assays are 0.182 μ M and 0.852 μ M, respectively. CP1 showed IC50s of 0.71 μ M and 0.672 μ M, respectively in the same assays.

“5. *The manuscript would benefit from editing for English language usage.*”

Response:

We thank for reviewer’s comments and suggestions. We improved the language and presentation of the manuscript.

Reference

1. Kaneda A, *et al.* The novel potent TEAD inhibitor, K-975, inhibits YAP1/TAZ-TEAD protein-protein interactions and exerts an anti-tumor effect on malignant pleural mesothelioma. *Am J Cancer Res* **10**, 4399-4415 (2020).
2. Holden JK, *et al.* Small Molecule Dysregulation of TEAD Lipidation Induces a Dominant-Negative Inhibition of Hippo Pathway Signaling. *Cell Rep* **31**, 107809 (2020).
3. Nouri K, *et al.* Identification of Celastrol as a Novel YAP-TEAD Inhibitor for Cancer Therapy by High Throughput Screening with Ultrasensitive YAP/TAZ-TEAD Biosensors. *Cancers (Basel)* **11**, (2019).
4. Bum-Erdene K, *et al.* Small-Molecule Covalent Modification of Conserved Cysteine Leads to Allosteric Inhibition of the TEADYap Protein-Protein Interaction. *Cell Chem Biol* **26**, 378-389 e313 (2019).
5. Tang TT, *et al.* Small Molecule Inhibitors of TEAD Auto-palmitoylation Selectively Inhibit Proliferation and Tumor Growth of NF2-deficient Mesothelioma. *Mol Cancer Ther* **20**, 986-998 (2021).
6. Zanconato F, *et al.* Genome-wide association between YAP/TAZ/TEAD and AP-1 at enhancers drives oncogenic growth. *Nat Cell Biol* **17**, 1218-1227 (2015).
7. Liu X, *et al.* Tead and AP1 Coordinate Transcription and Motility. *Cell reports* **14**, 1169-1180 (2016).
8. Jiao S, *et al.* VGLL4 targets a TCF4-TEAD4 complex to coregulate Wnt and Hippo signalling in colorectal cancer. *Nature communications* **8**, 14058 (2017).

9. Lee DH, *et al.* LATS-YAP/TAZ controls lineage specification by regulating TGFbeta signaling and Hnf4alpha expression during liver development. *Nature communications* **7**, 11961 (2016).
10. Beyer TA, *et al.* Switch enhancers interpret TGF-beta and Hippo signaling to control cell fate in human embryonic stem cells. *Cell reports* **5**, 1611-1624 (2013).
11. Pobbati AV, *et al.* Identification of Quinolinols as Activators of TEAD-Dependent Transcription. *ACS Chem Biol* **14**, 2909-2921 (2019).

Reviewers' Comments:

Reviewer #1:

Remarks to the Author:

The authors provided compelling responses to all my concerns, and I am now eager to support this MS for publication, although only after a minor correction:

Supplementary Figure 6a (and not Supplementary Figure 5a as the authors stated in their response to my point 2) is supposed to depict the TEAD binding sites around the PIK3C2B gene. Indeed, the right panel of that figure shows the presence of a TEAD4 peak upstream of that gene in multiple cell lines; instead, the left panel of Supplementary Figure 6a only shows a depiction of the gene and the writing "GSE66081 MDA-MB-231". I suppose that the authors meant to show there the ChIP-seq data from that dataset, but they somehow forgot to do that. Please correct it or remove the left panel.

Reviewer #3:

Remarks to the Author:

No comments necessary

Reviewer #4:

Remarks to the Author:

The authors have significantly revised the manuscript and have satisfactorily responded to the critiques raised in my original review. The additional Supplementary figures as well as additional text in the Discussion are helpful.

However, Please NOTE: Panels in Figure 2 are mislabeled! There is no panel "c" in the figure. The authors need to revise this Figure so that the panels are correctly labeled here, as well as in the revised text.

Response to Reviewers' comments

We appreciate the comments and suggestions from the reviewers. We have addressed all of these concerns.

Reviewer #1 (Remarks to the Author):

The authors provided compelling responses to all my concerns, and I am now eager to support this MS for publication, although only after a minor correction:

Supplementary Figure 6a (and not Supplementary Figure 5a as the authors stated in their response to my point 2) is supposed to depict the TEAD binding sites around the PIK3C2B gene. Indeed, the right panel of that figure shows the presence of a TEAD4 peak upstream of that gene in multiple cell lines; instead, the left panel of Supplementary Figure 6a only shows a depiction of the gene and the writing "GSE66081 MDA-MB-231". I suppose that the authors meant to show there the CHIP-seq data from that dataset, but they somehow forgot to do that. Please correct it or remove the left panel.

Response: Thanks for reviewer's suggestion. As suggested, we have removed the left panel of supplementary Figure 6a.

Reviewer #3 (Remarks to the Author):

No comments necessary

Reviewer #4 (Remarks to the Author):

The authors have significantly revised the manuscript and have satisfactorily responded to the critiques raised in my original review. The additional Supplementary figures as well as additional text in the Discussion are helpful.

However, Please NOTE: Panels in Figure 2 are mislabeled! There is no panel "c" in the figure. The authors need to revise this Figure so that the panels are correctly labeled here, as well as in the revised text.

Response: Thanks a lot for reviewer's comments. We have revised the Figure 2 and confirmed all the panels in Figure 2 are correctly labeled.